# Robust Neural Posterior Estimation and Statistical Model Criticism

**Daniel Ward**[1]  **Patrick Cannon**[2]  **Mark Beaumont**[3]  **Matteo Fasiolo**[1]
**Sebastian M. Schmon**[2,4]

[1]School of Mathematics, Bristol University, UK
[2]Improbable, UK
[3]School of Biological Sciences, Bristol University, UK
[4]Department of Mathematical Sciences, Durham University, UK

## Abstract

Computer simulations have proven a valuable tool for understanding complex phenomena across the sciences. However, the utility of simulators for modelling and forecasting purposes is often restricted by low data quality, as well as practical limits to model fidelity. In order to circumvent these difficulties, we argue that modellers must treat simulators as idealistic representations of the true data generating process, and consequently should thoughtfully consider the risk of *model misspecification*. In this work we revisit neural posterior estimation (NPE), a class of algorithms that enable black-box parameter inference in simulation models, and consider the implication of a simulation-to-reality gap. While recent works have demonstrated reliable performance of these methods, the analyses have been performed using synthetic data generated by the simulator model itself, and have therefore only addressed the well-specified case. In this paper, we find that the presence of misspecification, in contrast, leads to unreliable inference when NPE is used naïvely. As a remedy we argue that principled scientific inquiry with simulators should incorporate a *model criticism* component, to facilitate interpretable identification of misspecification and a *robust inference* component, to fit 'wrong but useful' models. We propose robust neural posterior estimation (RNPE), an extension of NPE to simultaneously achieve both these aims, through explicitly modelling the discrepancies between simulations and the observed data. We assess the approach on a range of artificially misspecified examples, and find RNPE performs well across the tasks, whereas naïvely using NPE leads to misleading and erratic posteriors.

## 1  Introduction

Stochastic simulators have become a ubiquitous modelling tool across the sciences and are regularly applied to some of the most complex and challenging problems of scientific interest, including climate change (see e.g. Randall et al., 2007), particle physics (e.g. Brehmer et al., 2020), and the Covid-19 pandemic (e.g. Ferguson et al., 2020). Simulators implicitly define a likelihood function $p(\boldsymbol{x} \mid \boldsymbol{\theta})$, where $\boldsymbol{x}$ is the simulator output and $\boldsymbol{\theta}$ are the simulator parameters. Although running the simulator to sample from the model is straightforward, the inherent complexity of simulators often makes analytic calculation of the likelihood intractable. As a result, classical inference techniques to find the parameter posterior $p(\boldsymbol{\theta} \mid \boldsymbol{x})$ such as Markov chain Monte Carlo (MCMC) (Metropolis et al., 1953) are infeasible. To overcome this issue, a large family of simulation-based inference (SBI) methods have been developed that allow parameter inference to be performed on arbitrary black-box simulators (see Cranmer et al., 2020). Broadly, these approaches estimate a function that allows access to an

36th Conference on Neural Information Processing Systems (NeurIPS 2022).

approximate posterior. This can be achieved by approximating the posterior directly (Papamakarios and Murray, 2016; Greenberg et al., 2019; Lueckmann et al., 2017), or indirectly, via approximating the likelihood (Papamakarios et al., 2019b) or likelihood-to-evidence ratio (Hermans et al., 2020; Thomas et al., 2022), from which the posterior can be sampled using MCMC. For estimating the posterior or likelihood, neural posterior estimation (NPE) and neural likelihood estimation (NLE) have shown to be powerful approaches (Lueckmann et al., 2021), which rely on neural network-based conditional density estimators, such as normalising flows, to approximate the likelihood or posterior (Papamakarios et al., 2019a).

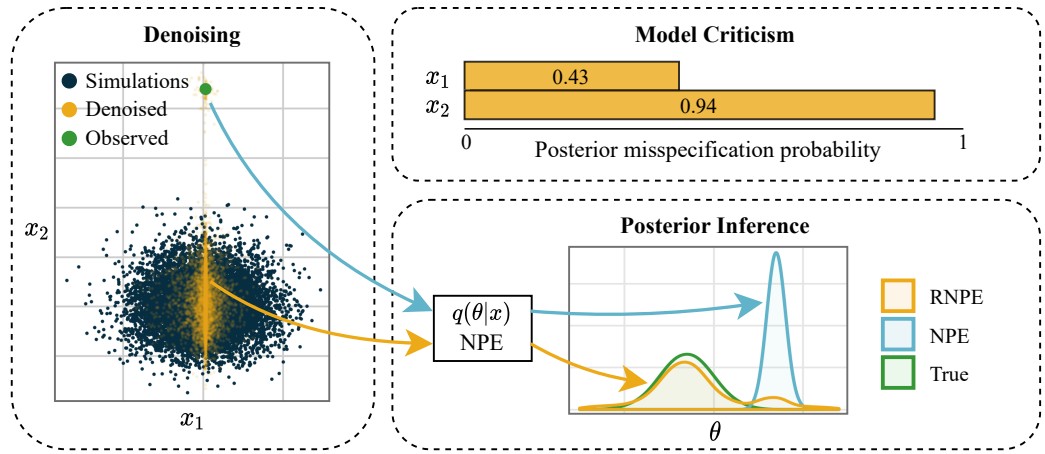

Figure 1: Overview of the robust neural posterior estimation (RNPE) framework. Through 'denoising' the observed data, we can simultaneously perform *model criticism* to identify how the model is misspecified, and perform inference *robust* to model misspecification. See Gaussian example in section 4.1 for details of the experiment.

To define misspecification, it is necessary to make a clear distinction between the true data generating process and the simulator: we will write $\boldsymbol{y}_o$ for observed data, following some unknown data generating process with distribution denoted $p^*$, and we will write $\boldsymbol{x} \sim p(\boldsymbol{x} \mid \boldsymbol{\theta})$ to denote samples from the simulator. In many applications of SBI, summary statistics are used to reduce the dimensionality of the problem. If this is the case, we use $\boldsymbol{x}$ and $\boldsymbol{y}_o$ to denote the simulated and observed summary statistics, rather than the raw data, and treat the raw data as an unobserved latent variable of the model. The simulator is said to be *misspecified* if the true data generating process does not fall within the family of distributions defined by the simulator, i.e. $p^* \notin \{p(\boldsymbol{x} \mid \boldsymbol{\theta}); \boldsymbol{\theta} \in \boldsymbol{\Theta}\}$. Simulators generally form idealised and simplistic representations of a more complex data generating process, and hence – like all statistical models – will be wrong to some extent. This misspecification often results in a discrepancy between simulated and observed data, sometimes termed a "simulation-to-reality gap" (e.g. Miglino et al., 1995). Despite this, the general approach in SBI is to learn a posterior approximation $q(\boldsymbol{\theta} \mid \boldsymbol{x})$ (either directly, or indirectly), using simulated data, $(\boldsymbol{x}, \boldsymbol{\theta}) \sim p(\boldsymbol{x} \mid \boldsymbol{\theta})p(\boldsymbol{\theta})$, and to condition on the observed data, $q(\boldsymbol{\theta} \mid \boldsymbol{x} = \boldsymbol{y}_o)$. This approach is only theoretically justified under the assumption that the simulator is well-specified; relying on this approach for misspecified simulators leads to two issues:

(i) As the posterior approximation $q(\boldsymbol{\theta} \mid \boldsymbol{x})$ is learned using samples from the simulator, we can only expect it to form a reasonable approximation of $p(\boldsymbol{\theta} \mid \boldsymbol{x})$ for regions well covered by simulations. For misspecified simulators, the observed data $\boldsymbol{y}_o$ may be very unlikely in $p(\boldsymbol{x}) = \int p(\boldsymbol{x}|\boldsymbol{\theta})p(\boldsymbol{\theta})d\boldsymbol{\theta}$, in which case the approximation $q(\boldsymbol{\theta} \mid \boldsymbol{x} = \boldsymbol{y}_o)$ often becomes poor (Cannon et al., 2022). In some cases, $\boldsymbol{y}_o$ may even fall outside the support of $p(\boldsymbol{x})$, which results in NPE attempting to extrapolate to estimate the undefined posterior.

(ii) It has been demonstrated more generally that Bayesian inference is frequently problematic under misspecification. Even for simple models, misspecification can lead to posterior concentration around "bad" models (Grünwald and Langford, 2007; Grünwald and Van Ommen, 2017) and poorly calibrated credible regions (Kleijn and van der Vaart, 2012; Syring and

Martin, 2019). These issues will persist even in the unlikely case that $q(\boldsymbol{\theta} \mid \boldsymbol{x})$ perfectly approximates $p(\boldsymbol{\theta} \mid \boldsymbol{x})$.

One approach to improve robustness is to incorporate an error model to explicitly model potential discrepancies between the observed data and simulations. A simple method would be to directly incorporate the error model within the simulator itself, such as by applying additive noise to the output to ensure that the observed data has reasonable marginal likelihood under the model (Cranmer et al., 2020). However, this approach has some limitations: *i)* it is challenging to interpret how the model is misspecified – an obvious choice is to use posterior predictive checks, but poor predictive performance could result from failed inference, limitations of simulator, or both; *ii)* retraining of the approximate posterior would be required to investigate different error models, which can be computationally expensive; and *iii)* the error model itself is likely tractable, which can be utilised to improve inference.

**Our contribution.** We propose robust neural posterior estimation (RNPE), an SBI framework that incorporates an error model, $p(\boldsymbol{y} \mid \boldsymbol{x})$ to account for discrepancies between simulations and the observed data, whilst avoiding the limitations of naïvely incorporating it directly into the simulator. We explicitly learn to invert the error model for the observed data prior to inference of the parameter posterior, a process we call 'denoising'. This approach yields the following advantages:

**Efficient Model Criticism.** Understanding a model's limitations is crucial for principled model development (Box, 1980; Gelman and Shalizi, 2013). By probabilistically inverting the error process for the observed data, and inferring any associated error model parameters, we can detect and interpret discrepancies between the simulated and observed data, allowing criticism of the simulator. This process can be achieved in a manner decoupled from inference, which provides efficiency benefits, whilst avoiding confounding inference failures with problems arising due to limitations of the simulator.

**Robust Inference.** Even after several rounds of model development and model criticism, the best available model will still frequently be misspecified to an appreciable degree. However, it is often still desirable to use 'wrong but useful' models for parameter inference and predictions, and hence the inference process should be robust to misspecification. By exploiting the amortisation[1] property of NPE, we can ensemble posterior estimates over a set of 'denoised' observations, leading to posterior estimates robust to misspecification.

An overview of the RNPE framework can be seen in Fig. 1. We examine the practical value of this approach for *model criticism* and *robust inference* on examples in which we artificially introduce realistic levels of misspecification.

## 2 Method

### 2.1 Error Model

As outlined above, we can explicitly model discrepancies between the observed data and simulations by introducing an error model $p(\boldsymbol{y} \mid \boldsymbol{x})$, such that the assumed generative model is

$$p(\boldsymbol{\theta}, \boldsymbol{x}, \boldsymbol{y}) = p(\boldsymbol{y} \mid \boldsymbol{x})p(\boldsymbol{x} \mid \boldsymbol{\theta})p(\boldsymbol{\theta}), \tag{1}$$

where we assume that the error is independent of the simulator parameters $\boldsymbol{\theta}$, and we treat $\boldsymbol{x}$ as an unobserved latent variable of the model. Using the equivalent factorisation

$$p(\boldsymbol{\theta}, \boldsymbol{x}, \boldsymbol{y}) = p(\boldsymbol{\theta} \mid \boldsymbol{x})p(\boldsymbol{x} \mid \boldsymbol{y})p(\boldsymbol{y}),$$

we can marginalise over the latent $\boldsymbol{x}$ and find an expression for the parameter posterior

$$p(\boldsymbol{\theta}, \boldsymbol{y}) = p(\boldsymbol{y}) \int p(\boldsymbol{x} \mid \boldsymbol{y})p(\boldsymbol{\theta} \mid \boldsymbol{x})\,\mathrm{d}\boldsymbol{x}, \tag{2}$$

$$p(\boldsymbol{\theta} \mid \boldsymbol{y}) = \int p(\boldsymbol{x} \mid \boldsymbol{y})p(\boldsymbol{\theta} \mid \boldsymbol{x})\,\mathrm{d}\boldsymbol{x} = \mathbb{E}_{\boldsymbol{x} \sim p(\boldsymbol{x}|\boldsymbol{y})}[p(\boldsymbol{\theta} \mid \boldsymbol{x})]. \tag{3}$$

Equation (3), implies that if we had access to $p(\boldsymbol{x} \mid \boldsymbol{y})$ and $p(\boldsymbol{\theta} \mid \boldsymbol{x})$, we could sample from the parameter posterior by first sampling from the posterior over the latent variables $\boldsymbol{x} \sim p(\boldsymbol{x} \mid \boldsymbol{y})$, and

---

[1]Amortisation refers to the ability of a conditional distribution approximation to condition on arbitrary instances of the conditioning variable, rather than being specialised to a particular instance.

then sampling from $p(\boldsymbol{\theta} \mid \boldsymbol{x})$. This approach reverses the data generating process from Equation (1), whilst propagating uncertainty throughout, first removing the error from the observed data by denoising, and then finding the associated simulator parameters given the denoised data. A Monte Carlo approximation of the expectation in Equation (3) can also be used to estimate the posterior density $p(\boldsymbol{\theta} \mid \boldsymbol{y})$, i.e.

$$p(\boldsymbol{\theta} \mid \boldsymbol{y}) \approx \frac{1}{M} \sum_{m=1}^{M} p(\boldsymbol{\theta} \mid \tilde{\boldsymbol{x}}_m), \qquad \tilde{\boldsymbol{x}}_1, \ldots, \tilde{\boldsymbol{x}}_M, \overset{\text{iid}}{\sim} p(\boldsymbol{x} \mid \boldsymbol{y}), \qquad (4)$$

however, we note that in some scenarios this approximation could have high variance. In practice, $p(\boldsymbol{x} \mid \boldsymbol{y})$ and $p(\boldsymbol{\theta} \mid \boldsymbol{x})$ are unknown. We can approximate $p(\boldsymbol{\theta} \mid \boldsymbol{x})$ by training a normalising flow $q(\boldsymbol{\theta} \mid \boldsymbol{x})$ on simulated data $(\boldsymbol{x}, \boldsymbol{\theta}) \sim p(\boldsymbol{x} \mid \boldsymbol{\theta})p(\boldsymbol{\theta})$, as is commonly done for NPE. To approximately sample from $p(\boldsymbol{x} \mid \boldsymbol{y})$, we *i)* specify an error model $p(\boldsymbol{y} \mid \boldsymbol{x})$, *ii)* train a normalising flow, $q(\boldsymbol{x})$, fitted to samples from the prior predictive distribution of the simulator $\boldsymbol{x} \sim p(\boldsymbol{x})$, and *iii)* sample $\tilde{\boldsymbol{x}} \sim p(\boldsymbol{x} \mid \boldsymbol{y})$, along with any error model parameters, using MCMC, as $\hat{p}(\boldsymbol{x} \mid \boldsymbol{y}) \propto p(\boldsymbol{y} \mid \boldsymbol{x})q(\boldsymbol{x})$. We denote samples from $\hat{p}(\boldsymbol{x} \mid \boldsymbol{y})$ and $q(\boldsymbol{\theta} \mid \boldsymbol{x})$ as $\tilde{\boldsymbol{x}}$ and $\tilde{\boldsymbol{\theta}}$ to avoid confusion with prior samples and simulations. Pseudo-code for the overall approach is given in Algorithm 1.

---

**Algorithm 1:** Robust neural posterior estimation (RNPE)

---

**for** $i$ *in* $1 : N$ **do**
1      Sample $\boldsymbol{\theta}_i \sim p(\boldsymbol{\theta})$
2      Simulate $\boldsymbol{x}_i \sim p(\boldsymbol{x} \mid \boldsymbol{\theta})$
**end**
3 Train NPE $q(\boldsymbol{\theta} \mid \boldsymbol{x})$ on $\{(\boldsymbol{\theta}_i, \boldsymbol{x}_i)\}_{i=1}^{N}$
4 Train $q(\boldsymbol{x})$ on $\{\boldsymbol{x}_i\}_{i=1}^{N}$
5 Sample $\tilde{\boldsymbol{x}}_m \sim \hat{p}(\boldsymbol{x} \mid \boldsymbol{y}_o) \propto p(\boldsymbol{y}_o \mid \boldsymbol{x})q(\boldsymbol{x}), \; m = 1, \ldots, M$ using MCMC
6 Sample $\tilde{\boldsymbol{\theta}}_m \sim q(\boldsymbol{\theta} \mid \tilde{\boldsymbol{x}}_m), \; m = 1, \ldots, M$
**return** $\{(\tilde{\boldsymbol{\theta}}_m, \tilde{\boldsymbol{x}}_m)\}_{m=1}^{M}$, samples drawn approximately from $p(\boldsymbol{\theta}, \boldsymbol{x} \mid \boldsymbol{y}_o)$

---

Under this framework, the standard SBI approach can be retrieved as the special case in which $p(\boldsymbol{y} \mid \boldsymbol{x}) = p(\boldsymbol{x} \mid \boldsymbol{y}) = \delta(\boldsymbol{x} - \boldsymbol{y})$, where $\delta$ is the Dirac delta distribution, meaning that we revert to assuming no discrepancy between the simulator and data generating process.

The idealised nature of simulation models implies that frequently there will be some characteristics of the observed data that are well captured by simulations, and some aspects in which there exists a discrepancy. Using a set of summary statistics for inference, rather than the raw data naturally captures an interpretable, low dimensional and diverse set of characteristics of the simulated and observed data. Because of this, we find that in general, reducing the raw data to summary statistics is advantageous for *model criticism*, as the practitioner can choose summary statistics which align with their belief about important model characteristics, and easily interpret any discrepancies (see Section 5). The requirement for domain specific knowledge to develop summary statistics has been alleviated by recent work (e.g. Fearnhead and Prangle, 2012; Chan et al., 2018; Chen et al., 2020), which seek to automatically embed the raw data to summary statistics; however, these embeddings often lack a tangible interpretation. Additionally, the embedding methods themselves may not be robust to misspecification. One approach is to use both hand-crafted and automatic summary statistics in tandem. Hereafter, however, we generally perform *model criticism* and *inference* using hand-crafted summary statistics, such that $\boldsymbol{x}$ and $\boldsymbol{y}_o$ refer to the simulated and observed summary statistics, *not* the raw data.

## 2.2   Spike and Slab

In practice, the true error model is unknown, and hence the error model should describe our prior belief over the discrepancy between the data generating process and simulations. Additionally, to facilitate model criticism, it should ideally allow easy assessment of which summary statistics are approximately well-specified and which are misspecified. Inspired by sparsity-inducing priors used in Bayesian regression literature (George and McCulloch, 1993), we use a "spike and slab" error

model on each summary statistic

$$\boldsymbol{x} \sim q(\boldsymbol{x}), \tag{5}$$

$$z_j \sim \text{Bernoulli}(\rho), \qquad\qquad j = 1, \ldots, D \tag{6}$$

$$y_j \mid x_j, z_j \sim \begin{cases} N(x_j, \sigma^2), & \text{if } z_j = 0 \\ \text{Cauchy}(x_j, \tau), & \text{if } z_j = 1 \end{cases} \qquad j = 1, \ldots, D \tag{7}$$

where $\boldsymbol{x} \in \mathbb{R}^D$. As the summary statistics have varying scales, we standardise them to have mean zero and variance one. The Bernoulli variables indicate whether the $j$-th summary statistic is approximately well-specified; we use the probability of $\rho = 0.5$ to express the belief that it is equally likely a summary statistic will be misspecified or well-specified *a priori*. When a summary statistic is well-specified (i.e., $z_j = 0$), we take the error distribution to be a tight Gaussian (a spike) centred on $x_j$, choosing $\sigma = 0.01$ to enforce consistency between the simulator and the observed data for the $j$-th statistic. In contrast, in the misspecified case (i.e., $z_j = 1$), we take the error model to be the much wider and heavy tailed Cauchy distribution (a slab). Generally, we might expect misspecification to be relatively subtle, but some model inadequacies can lead to catastrophically misspecified summary statistics. We chose the Cauchy scale $\tau = 0.25$, to reflect this, as it places half the mass within $\pm 0.25$ standard deviations of $x_j$, but the long tails accommodate summary statistics that are highly misspecified. The sparsity-inducing effect of this error model matches what we consider to be a common scenario, namely that a proportion of the summary statistics are jointly consistent with the simulator, whereas others may be incompatible. If we expect *a priori* more or fewer of the summary statistics to be misspecified, the prior misspecification probability can be adjusted accordingly. By marginalising over $\boldsymbol{z}$, the spike and slab error model can also be written as

$$p(\boldsymbol{y} \mid \boldsymbol{x}) = \prod_{j=1}^{D} \left[ (1 - \rho) \cdot p(y_j \mid x_j, z_j = 0) + \rho \cdot p(y_j \mid x_j, z_j = 1) \right], \tag{8}$$

and as such can be viewed as an equally weighted mixture of the Gaussian spike, $p(y_j \mid x_j, z_j = 0)$, and the Cauchy slab, $p(y_j \mid x_j, z_j = 1)$, for each summary statistic. We note that error model hyperparameter tuning approaches could be considered, e.g. a reasonable heuristic would be to choose an error model that is broad enough that the denoised data $\tilde{\boldsymbol{x}}$ tend not to be outliers in $q(\boldsymbol{x})$, compared to simulations[2]. However, we chose to keep the hyperparameters consistent across tasks, to avoid the risk of overfitting to the tasks and to demonstrate that neither strong prior knowledge on the error model, nor careful hyperparameter tuning is necessary to yield substantial improvements in performance.

A key advantage of the spike and slab error model is given by the latent variable $\boldsymbol{z}$. Similar to posterior inclusion probabilities in Bayesian regression, the posterior frequency of being in the slab, $\Pr(z_j = 1 \mid \boldsymbol{y})$, can be used as an indicator of the *posterior misspecification probability* for the $j$-th summary statistic. By comparing to the prior misspecification probability $\rho$, we can say that if $\Pr(z_j = 1 \mid \boldsymbol{y}) > \rho$, the model provides evidence of misspecification for the $j$-th summary statistic, and if $\Pr(z_j = 1 \mid \boldsymbol{y}) < \rho$, it provides evidence it is well-specified (Talbott, 2016). For the purpose of generality, the latent variable $\boldsymbol{z}$ was not included when describing RNPE thus far. However, we can jointly infer the posterior $\hat{p}(\boldsymbol{x}, \boldsymbol{z} \mid \boldsymbol{y})$ in step 5 of algorithm 1, by using an MCMC algorithm that supports sampling both continuous and discrete variables. We used mixed Hamiltonian Monte Carlo (HMC) (Zhou, 2020, 2022) from the NumPyro python package (Phan et al., 2019), which is an adaptation of HMC (Neal et al., 2011; Duane et al., 1987), for this purpose.

## 3 Related Work

### 3.1 Model Criticism in SBI

Posterior predictive checks have been widely used in SBI, to compare the predictive samples to the observed data (e.g. Durkan et al., 2020; Greenberg et al., 2019; Papamakarios et al., 2019b). Although this is a form of model criticism, we note a key limitation of this approach is that if a discrepancy is discovered, it may not be clear whether this is due to the failure of the inference procedure, or due to simulator misspecification. In general, it may be possible to identify the presence of misspecification using anomaly/novelty detection. One such approach was suggested by Schmitt et al. (2021) in the

---

[2]This could for example be assessed by estimating highest density regions of $q(\boldsymbol{x})$, using the density quantile approach from Hyndman (1996).

context of SBI. However, this approach does not facilitate interpreting misspecification in the sense presented here, and it is not clear how to extend it to perform robust inference.

There are a few related methods that criticise a posterior estimate by assessing its calibration using simulated data (e.g. Hermans et al., 2021; Prangle et al., 2014; Talts et al., 2018); however, these approaches criticise the inference procedure, *not* the simulator itself, and the associated results rely on the simulator being well-specified.

## 3.2   Robust Inference in SBI

Approximate Bayesian computation (ABC) is a family of SBI methods characterised by their use of rejection sampling alongside a kernel function to compare the simulated and observed data (Tavaré et al., 1997; Pritchard et al., 1999; Beaumont et al., 2002). The simplest form of ABC samples candidate parameters from the prior distribution and accepts them with probability proportional to the similarity of the corresponding simulations to the observed data. The use of a kernel is often seen as a necessary evil, as enforcing an exact match between the observed and simulated data generally results in all samples being rejected. However, it has been noted that the kernel function can often be interpreted as specifying an error model distribution $p(\boldsymbol{y} \mid \boldsymbol{x})$ (Wilkinson, 2013). This idea has been expanded using generalised Bayesian inference, in which the kernel is replaced with a loss function, which can be chosen to use a robust measure of discrepancy such as the maximum mean discrepancy (MMD) (Chérief-Abdellatif and Alquier, 2020; Frazier et al., 2020a; Fujisawa et al., 2021; Park et al., 2015; Schmon et al., 2020; Dyer et al., 2021).

Synthetic likelihood is an SBI method that uses a Gaussian approximation of the likelihood (Wood, 2010). Frazier and Drovandi (2021) investigated a robust synthetic likelihood algorithm, in which the variance components of the Gaussian likelihood are expanded based on additional free parameters, inferred in a nested MCMC sampling scheme. In the context of the paper, they interpret this approach as "likelihood adjustment"; however, this could equivalently be reframed as the addition of an independent Gaussian error model, with a prior on the variances[3].

In contrast to more traditional SBI methods, few solutions have been proposed for performing robust inference with neural SBI methods. Perhaps the most common approach is to augment the model with additional nuisance parameters to expand the range of values the simulator can reasonably produce (Cranmer et al., 2020). This again can often be interpreted as a method of incorporating an error model within the simulator. However, as previously discussed, the primary limitation of this approach is that it does not easily facilitate *model criticism*.

## 4   Experimental setup

On three misspecified tasks, we benchmark the performance of three methods for parameter inference: *i)* RNPE, *ii)* noisy neural posterior estimation (NNPE) in which we directly approximate $p(\boldsymbol{\theta} \mid \boldsymbol{y})$ by incorporating the error model into the simulator, and *iii)* NPE, in which we assume no model error is present. To reliably assess performance, we repeat the inference procedure with 1000 different observations and ground truth parameter pairs for each misspecified task, and calculate two metrics: *i)* the log probability of the true parameter vector, $\boldsymbol{\theta}^*$, and *ii)* the posterior coverage properties (see Section 4.2). Further metrics, and results for the well-specified case, can be found in Appendix A. We further show examples of RNPE on a single observation for each task to demonstrate how RNPE can be applied in practice for model criticism and robust posterior inference.

For all experiments, we used $N = 50,000$ simulations, with $M = 100,000$ MCMC samples following 20,000 warm up steps. The MCMC chains were initialised using a random simulation, and $z_j = 1$ for $j = 1, \ldots, D$. To build the approximation $q(\boldsymbol{x})$, we used block neural autoregressive flows (De Cao et al., 2020), as we do not require the ability to sample directly from $q(\boldsymbol{x})$. For the approximation of $q(\boldsymbol{\theta} \mid \boldsymbol{x})$, we used neural spline flows (Durkan et al., 2019). For all tasks the hyperparameters were kept consistent; information on hyperparameter choices can be found in Appendix C. The code required to reproduce all the results from this manuscript is available at https://github.com/danielward27/rnpe.

### 4.1   Tasks

Detailed descriptions of each task can be found in Appendix B.

---

[3]This results due the fact that the addition of two independent Gaussian variables yields another Gaussian random variable, with covariance equal to the sum of the individual covariance matrices.

**Gaussian.** A simple Gaussian example from Frazier et al. (2020b). The task involves predicting a single parameter $\mu$, using the sample mean and variance of 100 i.i.d. samples drawn from $N(\mu, 1)$, where 1 is the variance of the Gaussian distribution. To artificially produce misspecified observations, the 100 i.i.d. are instead drawn from $N(\mu, 2)$. Despite its simplicity, this task usefully demonstrates that issues with misspecification can arise even in the simplest of models.

**Gaussian Linear.** A 10-d linear Gaussian model from Lueckmann et al. (2021). The parameter $\boldsymbol{\theta}$ is the mean vector of a Gaussian likelihood. Additive Gaussian noise is used to introduce misspecification. For this task, the spike and slab error model employed will be reasonably misspecified, as the noise is not sparse or heavy tailed.

**SIR.** A stochastic model of epidemic spread with a time-varying infection rate. We attempt to infer two parameters, the infection rate $\beta$ and the recovery rate $\gamma$. We computed the following summary statistics: 1-3) `Mean`, `Median` and `Max` - the mean, median and maximum number of infections; 4) `Max Day` - the day of occurrence of the maximum number of infections, 5) `Half Day` - the day at which half the total number of infections was reached, and 6) `Autocor` - the mean autocorrelation (lag 1) of infections. To artificially produce misspecified observations, we chose to introduce a small degree of reporting delays, whereby weekend infection counts are reduced by 5%, which are subsequently recouped on the following Monday. Fig. 4d shows an example of a raw observation.

**CS.** A marked point process model of cancer and stromal (CS) cell development in 2D-space, in which the locality of a cell relative to unobserved parent points determines whether it is a cancer or a stromal cell. Similar point process models can be found in Jones-Todd et al. (2019). There are three Poisson rate parameters, corresponding to the cell rate $\lambda_c$, the parent rate $\lambda_p$ and the daughter rate $\lambda_d$, where $\lambda_d$ controls the number of nearest cells to each parent which become cancerous. The following summary statistics were used: 1-2) `N Cancer` and `N Stromal` - the number of cancer and stromal cells; 3-4) `Mean Min Dist` and `Max Min Dist` - the mean and maximum distance from stromal cells to their nearest cancer cell. To artificially produce misspecified observations, cancer cells were removed if they fell too close to a parent point, mimicking necrosis that often occurs in core regions of tumours. A plot of an example misspecified simulation and a corresponding artificially produced observation can be found in Fig. 9 in the appendix.

## 4.2 Performance Metrics

**Log probability of $\theta^*$.** The mean posterior log probability of the true parameters over multiple observations is an extensively used performance metric (e.g. Papamakarios and Murray, 2016; Hermans et al., 2020; Durkan et al., 2020), and is closely related to the average KL divergence between the true and approximate posteriors (Lueckmann et al., 2021). We found that NPE would occasionally fail catastrophically, placing negligible posterior mass on the true parameters, skewing the mean estimates. Due to this, we instead use box plots to visualise the distribution over the log-probabilities of the true parameters, rather than reporting the mean.

**Posterior Coverage.** Given a confidence level $\alpha$ and a posterior approximation $\hat{p}(\boldsymbol{\theta} \mid \boldsymbol{y})$, let $\mathrm{HDR}_{\hat{p}(\boldsymbol{\theta}|\boldsymbol{y})}(1 - \alpha)$ represent its $1 - \alpha$ highest posterior density region, i.e. the smallest region that contains at least $100(1 - \alpha)\%$ of the mass of $\hat{p}(\boldsymbol{\theta} \mid \boldsymbol{y})$. The expected posterior coverage is the frequency with which the true parameter value falls within this highest density region

$$\mathbb{E}_{\boldsymbol{\theta}^*, \boldsymbol{y} \sim p(\boldsymbol{\theta}, \boldsymbol{y})} \left[ \mathbb{1}\{\boldsymbol{\theta}^* \in \mathrm{HDR}_{\hat{p}(\boldsymbol{\theta}|\boldsymbol{y})}(1 - \alpha)\} \right] \tag{9}$$

where $\mathbb{1}$ is the indicator function. Posterior coverage is useful for assessing whether an inference procedure is likely to yield overconfident or conservative posterior estimates (Hermans et al., 2021). We empirically estimate this expectation, approximating the highest density regions using the density quantile approach outlined in Hyndman (1996), with 10,000 samples from the approximate posterior.

## 5 Results and Discussion

### 5.1 Overall Performance

The results shown in Fig. 2 show that both RNPE and NNPE perform better than NPE, both in terms of producing posteriors with more conservative coverage properties (Fig. 2a) and by tending to place more mass on $\boldsymbol{\theta}^*$ (Fig. 2b). It is important to note that having conservative posterior estimates is generally considered preferable to having overconfident posterior estimates, as the latter could lead to drawing of erroneous scientific conclusions. RNPE slightly outperforms NNPE in terms of the mass placed on $\boldsymbol{\theta}^*$. We hypothesise this increased performance is due to RNPE exploiting the fact that the

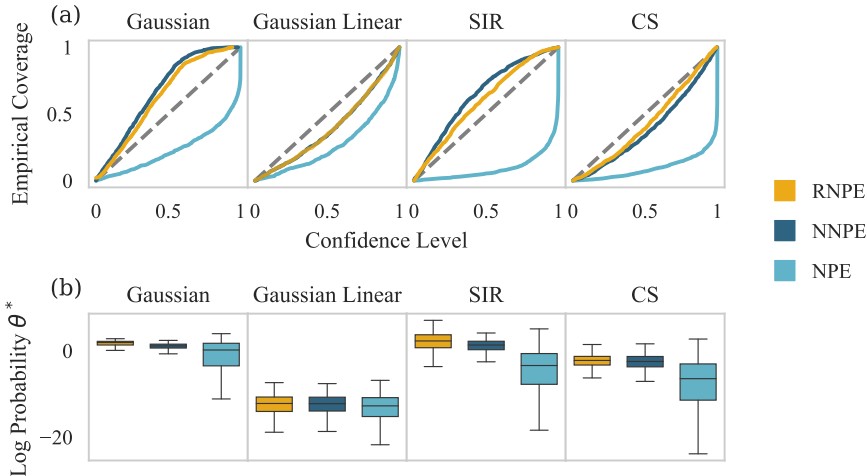

Figure 2: Comparative performance of RNPE, NNPE and NPE on the four misspecified tasks, using 1000 different observation and true parameter pairs. (a) The coverage of each approach at different confidence levels. A well-calibrated posterior follows the dotted line, with conservative posteriors lying above the dotted line, and overconfident posteriors falling below the dotted line. (b) The log-probability of the true parameters $\theta^*$ in the approximate posteriors.

error model is known and tractable, rather than naïvely relying on NPE to account for it. For all the tasks the error model is misspecified: these results highlight the importance of accounting for model misspecification, even if doing so may introduce a misspecified error model.

## 5.2 Examples

**Gaussian.** An example for the Gaussian task can be seen in Fig. 1, in which $x_1$ corresponds to the mean statistic, and $x_2$ the variance. From observing the posterior misspecification probabilities and the denoised samples, the results clearly identify a discrepancy in which the observed variance statistic is higher than the simulated variance statistics, allowing identification of source of misspecification. As this is a simple example, we can analytically access the true posterior (obtained using the true data generating model). RNPE produces a posterior close to the ground-truth, whereas NPE confidently places most of the posterior mass far from the ground-truth posterior.

**Gaussian Linear.** For the Gaussian Linear task, misspecification was introduced by applying additive Gaussian noise. Due to the lack of structure (e.g. correlations between simulation output dimensions), and the lack of sparsity in the errors, the model generally cannot provide much evidence for or against misspecification, as shown by Fig. 3. Corresponding pair plots of the denoised data and posterior are shown in Figs. 10-11, in the appendix.

**SIR.** In the SIR model, the observed data were artificially corrupted by mimicking reporting delays over the weekend. With this in mind, we can consider how the *model criticism* aspect of RNPE could facilitate identification of the issue. Fig. 4a shows that most summary statistics are more likely to be well-specified than was believed *a priori*, with the clear exception of the `Autocor` summary statistic, which is inferred to be misspecified. Fig. 4b shows a density plot of the denoised data for the `Autocor` summary statistic (for a full pair plot, see Fig. 12, in the appendix), showing that the observed autocorrelation is much lower than would be expected under the model. This may prompt the practitioner to investigate the

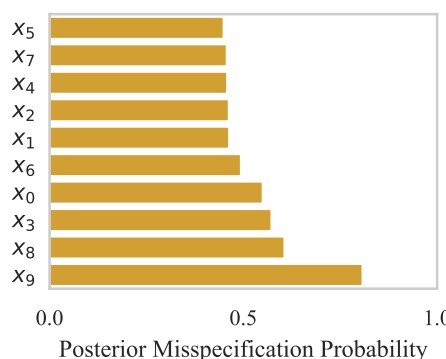

Figure 3: Posterior misspecification probabilities for an example of the Gaussian Linear task.

day-to-day changes in infections, in which case the discontinuities introduced by the reporting delays could be identified. Fig. 4c shows the posteriors for RNPE and NPE. RNPE yields a posterior with far lower variance than NPE, demonstrating that it does not always achieve robustness simply by producing broader posterior estimates.

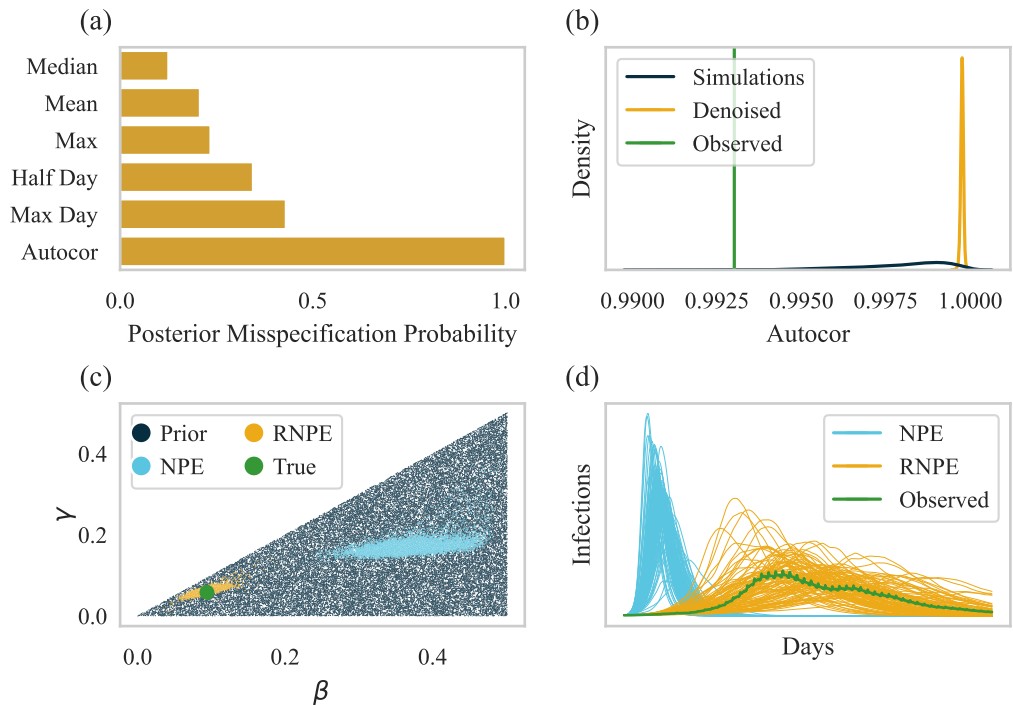

Figure 4: The results for an example of the SIR task. (a) The inferred posterior misspecification probabilities for each summary statistic. (b) Density plot for the denoised samples of the most misspecified summary statistic (autocorrelation). (c) The posteriors for both NPE and RNPE and; (d) The associated posterior predictive distributions for NPE and RNPE. Note the (artificially introduced) reporting delays are visible for the raw observed data.

**CS.** In the CS model, the observed data was corrupted by removing cells in the core regions of tumours, mimicking necrosis (see Fig. 9 in the appendix). This misspecification primarily impacts the `N Cancer` summary statistic (the number of cancer cells). From Figs. 5a,5b, we can see that the results provide evidence that the `N Cancer` statistic is misspecified, with the observed data having fewer cancer cells than would be expected if it was generated from the simulator. Once again, alongside domain knowledge, this information points the practitioner to the source of misspecification (necrosis), and hence the model could be adjusted appropriately. For this example, NPE produced an overconfident posterior with little mass on the true parameters, whereas RNPE does not exhibit this issue (Fig. 5c).

### 5.3 Indeterminacy

In some cases, there can be indeterminacy when inferring misspecification, in which it is impossible to determine which summary statistic is misspecified, despite clear evidence of misspecification. This occurs, for example, when a model can only be consistent with one observed summary statistic, e.g. $x_1$, at the expense of being able to recreate another summary statistic, e.g. $x_2$ (or vice versa). RNPE allows these trade-offs to be identified, as the statistics will show a small frequency of being jointly well-specified, $\Pr(z_1 = 0,\ z_2 = 0 \mid \boldsymbol{y})$, alongside reasonable probabilities of either statistic being well-specified, $\Pr(z_1 = 1, z_2 = 0 \mid \boldsymbol{y})$ and $\Pr(z_1 = 0,\ z_2 = 1 \mid \boldsymbol{y})$. We show an example of this in Fig. 8 in the Appendix.

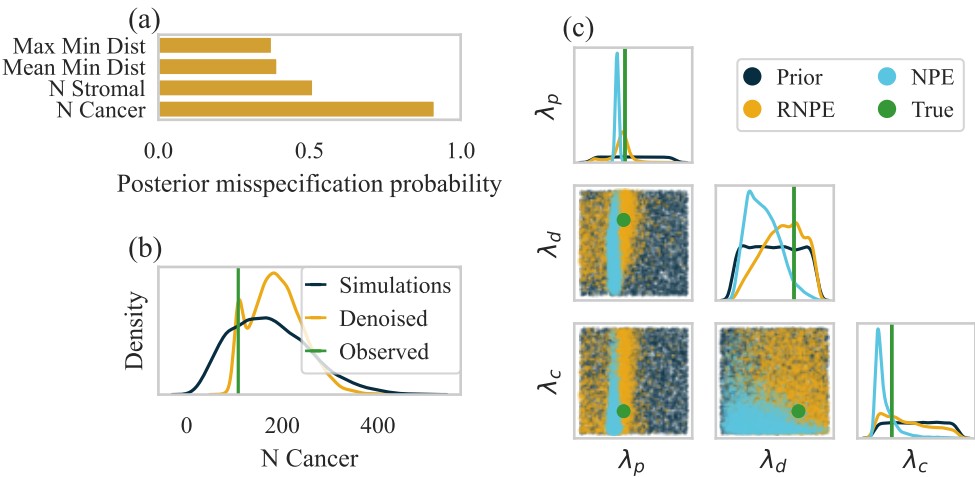

Figure 5: The results for an example of the CS task. (a) The inferred posterior misspecification probabilities for each summary statistic. (b) Density plot for the denoised samples of the `N Cancer` summary statistic. (c) A pair plot of the posteriors for both NPE and RNPE.

## 6 Conclusions

We described RNPE, an approach that can be applied to any black-box simulator to simultaneously perform *model criticism* and *robust inference*. RNPE is philosophically similar to existing robust SBI approaches for ABC and synthetic likelihood (Wilkinson, 2013; Frazier and Drovandi, 2021). However, this work introduces three novel contributions: *i)* we apply the idea of using error models for robust inference to a neural SBI method, which has advantages in simulation efficiency, flexibility and scaling to high dimensional problems (Lueckmann et al., 2021), *ii)* we facilitate model criticism *decoupled* from inference, which avoids confounding of inference failures and model misspecification, and *iii)* we use a spike-and-slab error model to obtain explicit misspecification probability diagnostics for summary statistics, to facilitate *model criticism*. Our work is not without limitations. Firstly, RNPE is more computationally costly than NPE, requiring an additional MCMC step and fitting of $q(\boldsymbol{x})$. Secondly, we only evaluated performance of a single choice of error model on a limited number of tasks. It would be beneficial in the future to have a larger set of realistic misspecified tasks for assessing performance, similar to the work by Lueckmann et al. (2021) for the well-specified case. Finally, we restricted our focus to NPE; in future work it would be interesting to consider other SBI algorithms and to compare results with existing methods for robust inference.

We demonstrated with four examples that RNPE consistently outperforms NPE for parameter inference under misspecification, tending to put more posterior mass on the true parameters whilst also exhibiting more conservative coverage properties. The tasks were designed to mimic levels of misspecification that could be realistically encountered, and hence the poor performance of NPE calls into question its use in practice, particularly when the level of misspecification could be significant. The substantial difference in performance highlights, as suggested by Box (1980), that we should not have blind faith that a model is sufficiently accurate, and therefore must devise methods for performing *model criticism* and *robust inference*. With further research and wider adoption of these principles in SBI, we believe simulators will only become more powerful tools for scientific discovery.

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
