# Appendix

## A   Additional Results

**C2ST**

In the classifier 2-sample test (Friedman, 2004; Lopez-Paz and Oquab, 2016) a classifier is trained to distinguish between a true and inferred posterior distribution. In the case that the classifier struggles to distinguish between the samples (i.e. when the posterior approximation is good), the C2ST score approaches $0.5$. In contrast, when the posterior approximation is poor the C2ST score will be close to $1$. Here we take the 'true' posterior to be the analytical posterior obtained under the true data generating process. For calculation of the C2ST metric we used a multilayer perceptron, as implemented in the sbibm python package (Lueckmann et al., 2021). This metric requires a tractable posterior, so is only available for the Gaussian and Gaussian Linear tasks. We note that the error model used in RNPE and NNPE is misspecified, so we cannot realistically expect a C2ST score close to $0.5$, but the results are useful to highlight that even a misspecified error model can improve performance. These results are shown in Fig. 6

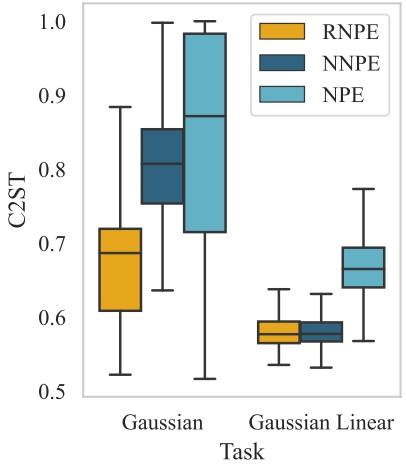

Figure 6: The C2ST accuracy, between the true posterior (under the true data generating process), and the approximate posteriors for each method.

**Mean Square Error**

In addition to calculating the log-probability of $\boldsymbol{\theta}^*$ and the posterior coverage properties (see Section 4.2 and Fig. 2), we also considered taking the posterior mean as a parameter point estimate and comparing this to the true parameters $\boldsymbol{\theta}^*$. Over the 1000 different true parameters and observation pairs, we compared the posterior means to the true parameters using the mean squared error

$$\text{MSE} = \frac{1}{1000} \sum_{i=1}^{1000} (\bar{\theta}_{ij} - \theta_{ij}^*)^2,$$

where $\bar{\theta}_{ij}$ is the posterior mean of the $j$-th parameter, resulting from either NPE or RNPE, and $\theta_{ij}^*$ is the ground truth parameter sampled from the prior. These results are shown in Table 1.

| Task | Parameter | Mean Squared Error | | |
|------|-----------|------|------|------|
| | | NNPE | NPE | RNPE |
| Gaussian | $\mu$ | 0.04 | 0.17 | 0.02 |
| Gaussian Linear | $\boldsymbol{\theta}$ | 0.70 | 0.75 | 0.70 |
| CS | $\lambda_c$ | 0.12 | 0.12 | 0.13 |
| | $\lambda_d$ | 1.21 | 1.92 | 1.21 |
| | $\lambda_p$ | 0.80 | 1.10 | 0.75 |
| SIR | $\beta$ | 0.20 | 1.10 | 0.13 |
| | $\gamma$ | 0.11 | 0.59 | 0.09 |

Table 1: The mean squared error between the posterior means and the true parameters for each task, estimated using 1000 different observation and true parameter pairs. The mean squared errors were calculated on standardised parameters, i.e. scaled based on the prior mean and variance, due to the differing scales of different parameters. For the Gaussian Linear task, we average the mean squared error across the parameters (rather than listing them for the 10 parameters individually).

**Well-Specified Case**

An obvious question to ask is, what is the impact of incorrectly using an error model when the simulator is well-specified? These results are shown in Fig. 7. In the well-specified case, NPE becomes well-calibrated (Fig. 7a), whereas RNPE produces slightly conservative posteriors. NPE performs marginally better in terms of the posterior mass placed on the true parameters (Fig. 7b). However, in the well-specified case, RNPE may indicate that the model is not likely to be misspecified, in which case the researcher could choose to collapse the error model by directly using $q(\boldsymbol{\theta} \mid \boldsymbol{x})$, with no additional cost.

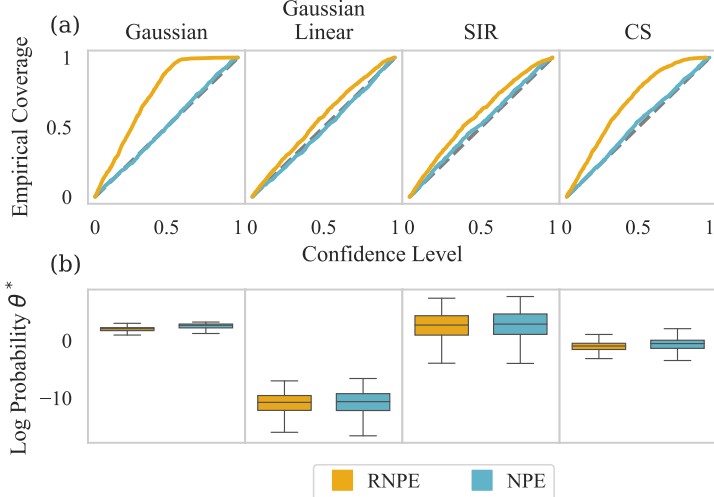

Figure 7: Comparative performance of RNPE and NPE on the four tasks without misspecification of the simulator, using 1000 different observation and true parameter pairs. (a) The coverage of each approach at different confidence levels. A well-calibrated posterior follows the dotted line, with conservative posteriors lying above the dotted line, and overconfident posteriors falling below the dotted line. (b) The log-probability of the true parameters $\boldsymbol{\theta}^*$ in the approximate posteriors.

## B Tasks

**Gaussian**

> **Model Description.** A simple Gaussian toy model used by Frazier et al. (2020b). The task has a single parameter $\mu$, which is the mean of a Gaussian distribution. A simulation consists of drawing 100 i.i.d. samples from $N(\mu, 1)$, which are summarised to the sample mean and variance.

> **Misspecification.** To artificially introduce misspecification, samples were drawn from $N(\mu, 2)$ for the observed data, rather than $N(\mu, 1)$, prior to summarising.

> **Priors.** $\mu \sim N(0, 25)$.

**Gaussian Linear**

> **Model Description.** A misspecified extension of a simple Gaussian toy model used by Lueckmann et al. (2021). The task involves inferring the mean vector $\boldsymbol{\theta}$ of a 10-d Gaussian distribution, where a simulation consists of sampling this distribution i.e. $\boldsymbol{x} \mid \boldsymbol{\theta} \sim N(\boldsymbol{\theta}, \ 0.1 \cdot \boldsymbol{I}_{10})$.

> **Misspecification.** To artificially introduce misspecification, we add Gaussian additive noise such that $\boldsymbol{y} \mid \boldsymbol{x} \sim N(\boldsymbol{x}, \ 0.1 \cdot \boldsymbol{I}_{10})$, or equivalently, we can view this as a misspecified variance in the simulator such that $\boldsymbol{y} \mid \boldsymbol{\theta} \sim N(\boldsymbol{\theta}, \ 0.2 \cdot \boldsymbol{I}_{10})$.

> **Priors.** $\boldsymbol{\theta} \sim N(0, \ 0.1 \cdot \boldsymbol{I}_{10})$.

## SIR

**Model Description.** The SIR model is an idealised model of disease spread, where a series of differential equations describe the rates of transitions between three states: susceptible ($s$), infected ($i$) and recovered ($r$). The simplest (and deterministic) model is

$$\frac{ds}{dt} = -\beta s i, \quad \frac{di}{dt} = \beta s i - \gamma i, \quad \frac{dr}{dt} = \gamma i, \tag{10}$$

where $\beta$ is the infection rate, and $\gamma$ the recovery rate. We use a stochastic extension of this model, with a time-varying infection rate $\tilde{\beta}_t$ (see Atkeson et al., 2020; Perla et al., 2022), reflecting for example policy changes or mutations in the virus. This process is parameterised using the basic reproduction number, $R_{0t} = \frac{\tilde{\beta}_t}{\gamma}$

$$dR_{0t} = \eta \left( \frac{\beta}{\gamma} - R_{0t} \right) dt + \sigma \sqrt{R_0} dW_t, \tag{11}$$

where $\eta$ is the mean reversion strength of $R_{0t}$ to $\frac{\beta}{\gamma}$, $\sigma$ is the volatility, and $W_t$ is Brownian motion. We set $\eta = 0.05$ and $\sigma = 0.05$ and focused on inferring $\beta$ and $\gamma$. This model was implemented in Julia, using the DifferentialEquations package (Rackauckas and Nie, 2017). We scaled the resulting infection trajectories by 100,000 (representing the population size), and used summary statistics of the daily infection counts over 365 days, as described in Section 4.1.

**Misspecification.** To artificially introduce misspecification, we reduce the infection counts on weekends by 5%, and these infections are recouped on the subsequent Monday, mimicking reporting delays over weekends. We (arbitrarily) take the first day to be Monday. An example of a simulations, compared to an observation with reporting delays is shown in Fig. 4.

**Priors.** We used $\beta \sim \text{Uniform}(0, 0.5)$ and $\gamma \sim \text{Uniform}(0, 0.5)$, but rejected samples where $\gamma > \beta$ as frequently the number of infections stayed near zero (as the recovery rate is greater than the infection rate).

## CS

**Model Description.** The CS model simulates cancer and stromal cell development on a 2D plane. The total number of cells $N^c$, number of unobserved parents $N^p$, and the number of daughter cells for each parent $N_i^d$ are sampled

$$N^c \sim \text{Poisson}(\lambda^c),$$
$$N^p \sim \text{Poisson}(\lambda^p),$$
$$N_i^d \sim \text{Poisson}(\lambda^d), \quad i = 1, \dots, N^p,$$

where $\lambda^c$, $\lambda^p$ and $\lambda^d$ are the three Poisson rate parameters we attempt to infer. The cell positions $\{c_i\}_{i=1}^{N^c}$ and parent positions $\{p_i\}_{i=1}^{N^p}$ are generated uniformly on the 2D plane (i.e. as two homogeneous point processes). Let $r_i$ represent the euclidean distance from parent $p_i$ to its $N_i^d$-th nearest cell. $r_i$ is the affected radius, such that cells are assigned to be cancerous if they fell on or within this distance of $p_i$. We used summary statistics based on cell type counts and distances, as described in Section 4.1. In order to efficiently calculate the distances, we made use of the Numba just-in-time compiler for Python (Lam et al., 2015). The distance based metrics (`Mean Min Dist` and `Max Min Dist`) were approximated empirically through sampling 50 stromal cells, as the full distance matrix was too expensive to compute.

**Misspecification.** For each parent, we sampled $w_i \sim \text{Bernoulli}(0.75)$. If $w_i = 1$, cancer cells falling within $0.8 r_i$ of the corresponding parent $p_i$ were removed. An example of this is shown in Fig. 9. This mimics necrosis that often occurs in the centre regions of tumours (see e.g., Jones-Todd et al., 2019).

**Priors.** $\lambda_c \sim \text{Uniform}(200, 1500), \; \lambda_p \sim \text{Uniform}(3, 20), \; \lambda_d \sim \text{Uniform}(10, 20).$

## C   Hyperparameters

**Flows.** For $q(\boldsymbol{x})$ we used a block neural autoregressive flow (De Cao et al., 2020), with a single hidden layer of size $8D$, where $x \in \mathbb{R}^D$. For $q(\boldsymbol{\theta} \mid \boldsymbol{x})$ we used a neural spline flow (Durkan et al.,

2019), defining the transform on the interval $[-5, 5]$, using 10 spline segments, and 5 coupling layers. For both flows we used a standard Gaussian base distribution. We used custom normalising flow implementations using JAX (Bradbury et al., 2018) and Equinox (Kidger and Garcia, 2021), which can be found at `https://github.com/danielward27/flowjax`. We found JAX flow implementations alongside NumPyro (Phan et al., 2019) for performing HMC to be several orders of magnitude faster than relying on Pytorch implementations and Pyro (Bingham et al., 2019).

**Training.** For both flow approximations, we used a batch size of 256, using 10% of the data as a validation set. We trained for a maximum of 50 epochs, terminating training early if 5 consecutive epochs occurred with no improvement in the validation loss. For the block neural autoregressive flow $q(\boldsymbol{x})$, we used a learning rate of $1 \times 10^{-2}$, whereas for the neural spline flow $q(\boldsymbol{\theta} \mid \boldsymbol{x})$ we used a learning rate of $5 \times 10^{-4}$.

**MCMC.** We used 100,000 steps with 20,000 warm up steps. The trajectory length was set to 1, and the target acceptance probability to 0.95 to increase the robustness of the MCMC algorithm. All other hyperparameters were left to the defaults of the NumPyro Python package (Phan et al., 2019).

## D   Computational Resources

All experiments were performed on CPU, with a maximum of 8GB of RAM, using the computational facilities of the Advanced Computing Research Centre, University of Bristol - `http://www.bristol.ac.uk/acrc/`.

## E   Additional Figures

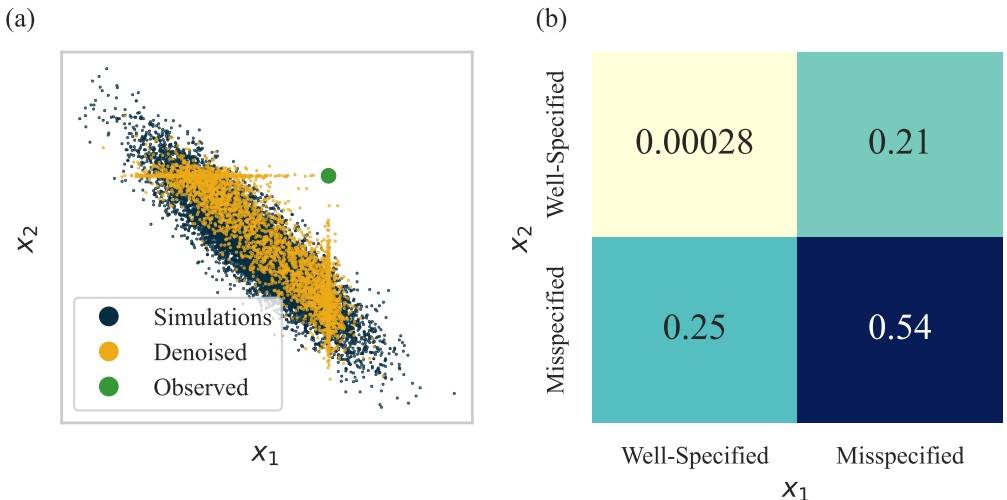

Figure 8: An example in which there is 'indeterminacy' in the inferred misspecification, where we take $p(\boldsymbol{x})$ to be a correlated Gaussian. (a) The denoised samples, compared to the observed data and simulations. (b) The frequencies with which the different pairwise combinations occurred in $\boldsymbol{z} \sim \hat{p}(\boldsymbol{z}|\boldsymbol{y})$, inferred during denoising. The results suggest a very low probability of both being well specified, despite reasonable probabilities of either $x_1$ or $x_2$ (exclusively) being well-specified, or both being misspecified. Despite the strong evidence for misspecification, the source of misspecification cannot be determined without further information.

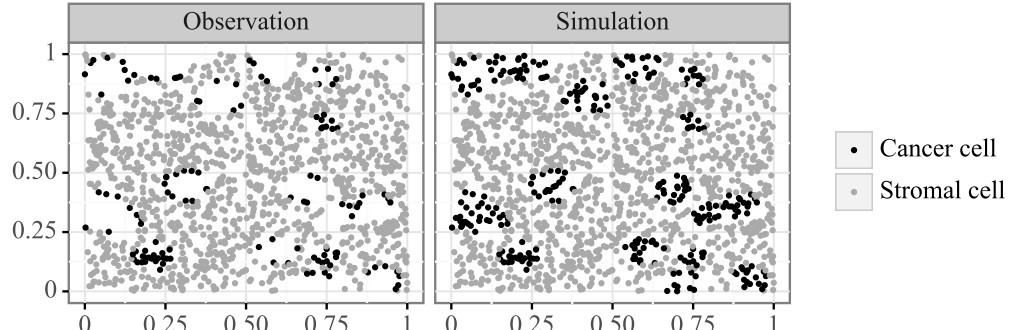

Figure 9: Example of a raw observation for the CS task, compared to the underlying simulation prior to corruption. In the core region of some tumours for the observed data, cancer cells have been removed, artificially mimicking necrosis. See Appendix B for more information.

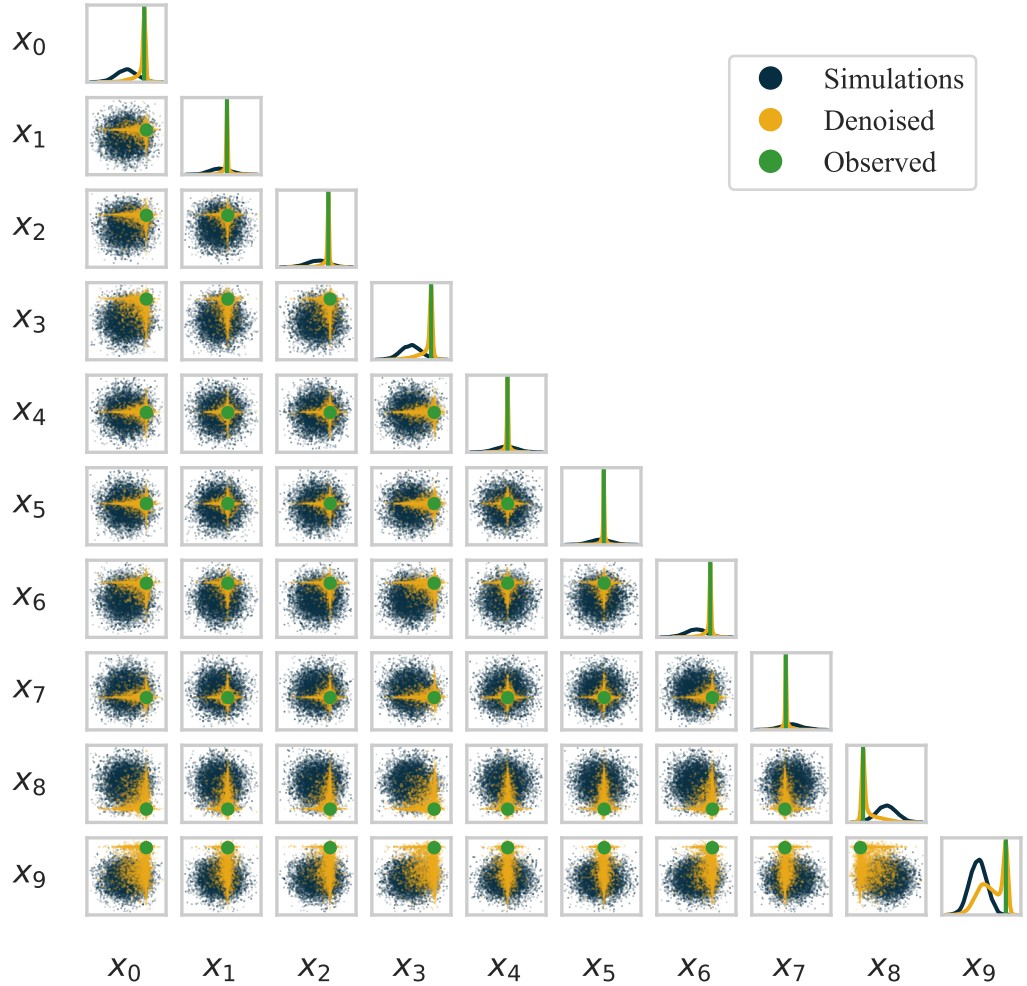

Figure 10: Pair plot of the denoised data for the Gaussian Linear task, corresponding to the example shown in Section 5.

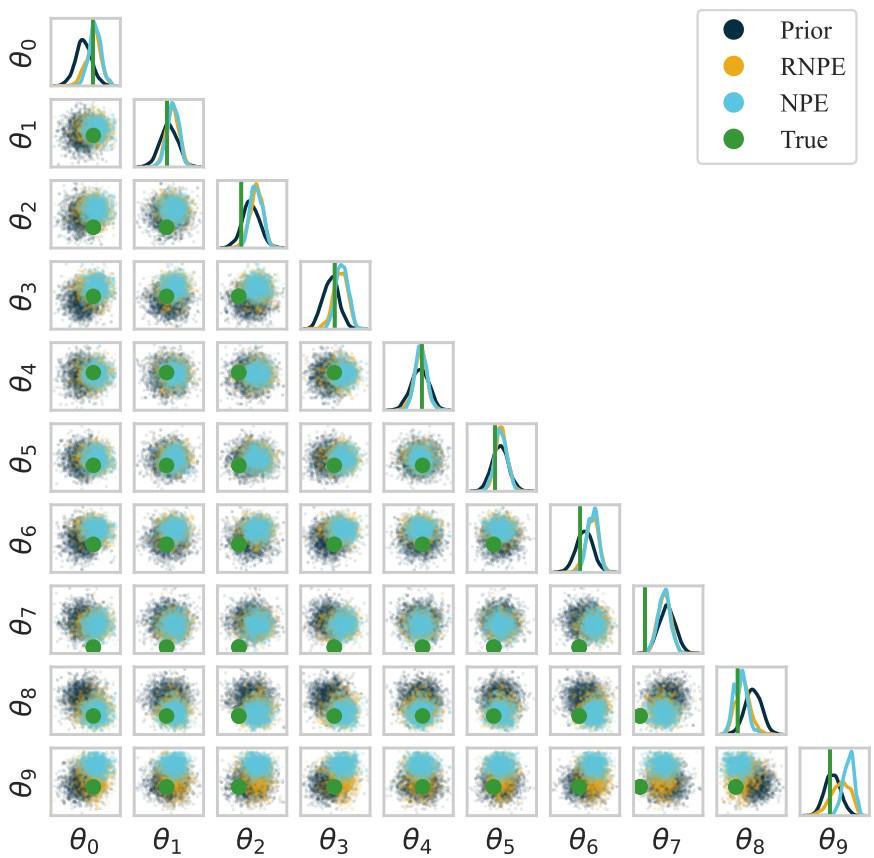

Figure 11: Pair plot of the posterior Gaussian Linear task, corresponding to the example shown in Section 5.

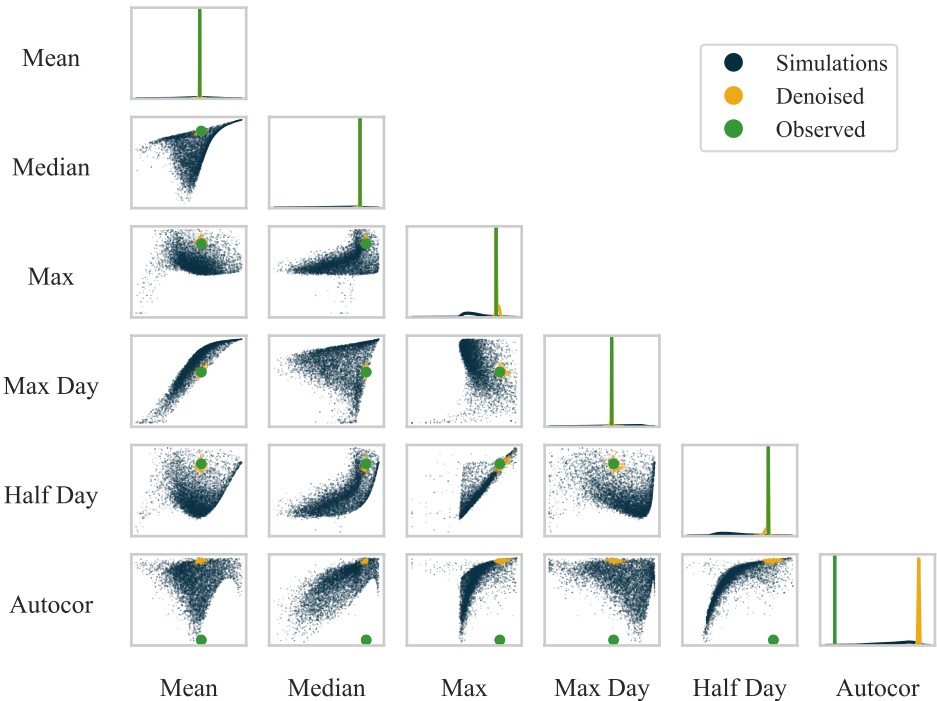

Figure 12: Pair plot of the denoised data for the SIR task, corresponding to the example shown in Section 5. Note that the observed data point obscures the denoised data along many axes, due to the shrinkage effect of the spike and slab error model. See Section 4.1 for the summary statistic definitions.

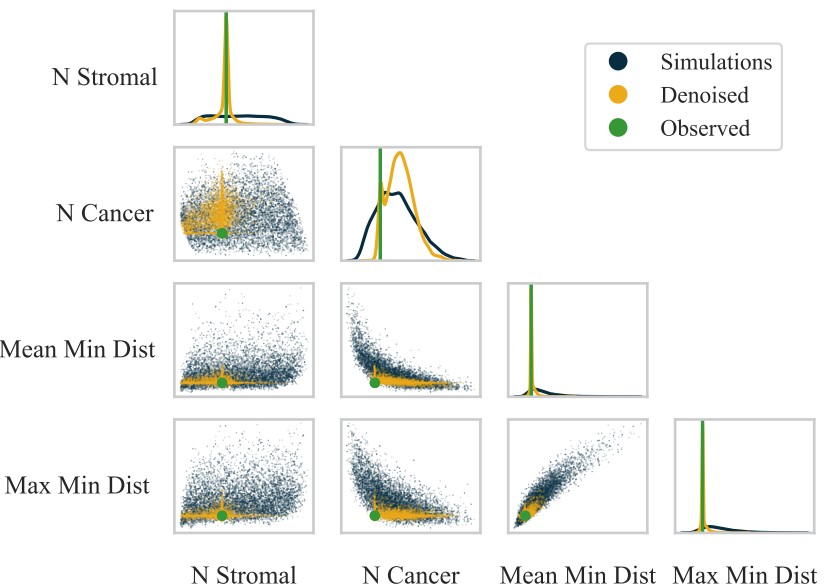

Figure 13: Pair plot of the denoised data for the CS task, corresponding to the example shown in Section 5. See Section 4.1 for the summary statistic definitions.
.

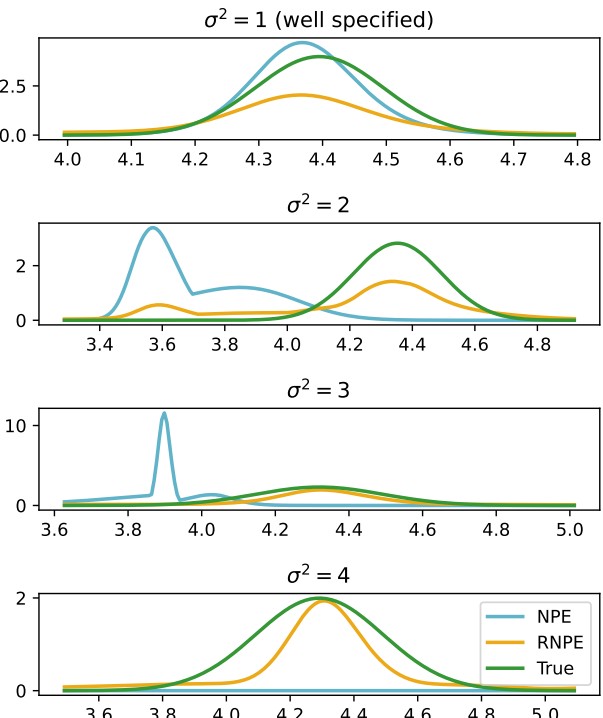

Figure 14: RNPE and NPE under increasing misspecification for the Gaussian task. Note that at $\sigma^2 = 4$, NPE failed to put significant mass within $\pm 4$ standard deviations of the true posterior mean. The "True" posterior is given by the posterior under the true data generating process model.