# OpenReview forum: "Robust Neural Posterior Estimation and Statistical Model Criticism"
_NeurIPS.cc/2022/Conference — NeurIPS 2022 Accept_

### Official Review · Reviewer_veSL · 2022-06-22

**Rating:** 3
**Confidence:** 2
**Soundness:** 3 good
**Presentation:** 3 good
**Contribution:** 1 poor

**Summary:**

The paper addresses the issue of model misspecification in simulation-based inference (sbi). Simulation-based inference refers to the task of inferring parameters of interest $\theta$ based on an observation $y_o$ and a simulator. The simulator is based on scientific knowledge and allows to sample synthetic observations $x \sim p(x|\theta)$. Sbi assumes that those simulations match the real data generation process. However, in practice, this simulator is imperfect and there is a gap between simulation and reality. This work aims to resolve this gap by introducing an error model $y \sim p(y|x)$ explicitly modeling the gap and proposing a strategy to perform robust inference using both the simulator and the error model. There are two contributions presented in this work:
* Perform robust inference with a neural sbi method by leveraging the error model
* get explicit misspecification posterior probabilities from the error model.

**Questions:**

As explained in the strengths and weaknesses section, my main questions would be:
* How does combining posterior misspecification with the simulator helps to perform model criticism?
* What advantages does it have over the naive approach of including the error model in the simulator for robust sbi?

**Limitations:**

I do not see any potential negative societal impact of this work.

**Strengths And Weaknesses:**

* (+) The paper is overall well written
* (+) Model misspecification is a hot topic in simulation-based inference
* (+) Experiments are convincing

* (-) My main concerns about this work are about the problem statement. The authors proposed to use two models for the inference: the simulator $p(x|\theta)$ and the error model $p(y|x)$ which is assumed to be known. In my opinion, if the error model were to be known a priori, it could be included in the simulator, hence viewing $x$ as a latent variable and $y$ as the simulated observable. In this case, classical simulation-based inference algorithms could be used. The big challenge with model misspecification is that, in practice, part of those misspecifications are unknown. One can then leverage real observations or assume specific types of misspecification without having a proper distribution and leverage Generalized Bayesian inference to be robust to those types of misspecification. Note that the authors do say something about this at l212-l216:
> In contrast to more traditional SBI methods, few solutions have been proposed for performing robust inference with neural SBI methods.  Perhaps the most common approach is to augment the model with additional nuisance parameters (e.g. additive noise), to expand the range of values the simulator can reasonably produce (Cranmer et al., 2020). This again can generally be interpreted as a naïve method of incorporating an error model and does not facilitate model criticism.
>
However, it is not clear to me how combining the two components provides additional value regarding the two contributions presented in the paper.
   - Contribution 1: perform robust inference with a neural sbi method. What advantages does it have over the naive approach of including the error model in the simulator?
   - Contribution 2: get explicit posterior misspecification probability. Why do we need to combine the error model with the simulator? Couldn't the error model be used alone for this task?

---

> ### Author Response · Authors · 2022-08-02
> **Thank you for your review.**
>
> > Contribution 1: perform robust inference with a neural sbi method. What advantages does it have over the naive approach of including the error model in the simulator?
>
> We appreciate this question. Adding noise directly to the simulator seems the most straightforward way to handle misspecification. However, using RNPE with an explicit error model as we do here has the following advantages:
>
> 1. Structure of the error model: Note that what we propose is a structural error model that allows for interpretation and critical assessment of model misspecification. Approaches using nuisance parameters commonly use a simpler error model, e.g. additive noise to extend the range of possible values.
> 2. Decoupling the simulator and the error model provides additional insight: By analysing differences between observed data $y$ and simulated data $x$ the modeller can be aided in improving the simulation model. Knowing which summary statistics are misspecified and observing the effect on the distribution can be crucial for model development.
> 3. Including what is known to improve computational efficiency: Since we do have a distributional model for the error we can include this information in the inference procedure. The alternative would be to rely on NPE to learn the dependence on a potentially difficult heavy tailed distribution (such as a Cauchy distribution), i.e. to recover an element of the simulator that is already known.
> 4. Black-box applicability: The error model can be integrated into existing black-box inference packages, such as SBI, and augment NPE. This allows practitioners access to interpretable insights into model misspecification without having to manipulate their code or alternatively to check a simulation model that is otherwise only available as black-box itself.
>
> It is evident that we could have been clearer when justifying the approach used, we will amend this in the final draft.
>
>
> >Contribution 2: get explicit posterior misspecification probability. Why do we need to combine the error model with the simulator? Couldn't the error model be used alone for this task?
>
> We would be grateful if you could provide further clarification on this remark. The posterior misspecification probabilities need to be based both on the simulator and noise process itself, because the misspecification probabilities are based on the discrepancy between the simulated and observed summary statistics.
>
> Let us know if you have any further questions, and please consider updating your score if you feel we have adequately addressed your concerns.

---

> > ### Comment · Reviewer_veSL · 2022-08-03
> > **Clarifications**
> >
> > I admit that my comment on contribution 2 was imprecise, let me clarify it. From my understanding, the goal is to compute $q(x, z | {y}\_{o}) \propto p({y}\_{o} | x, z) q(x) p(z)$. Indeed the simulator is used to compute $q(x)$ but never is $q(\theta | x)$ required. Computing those probabilities can thus be done without inverting the simulator.
> >
> > Therefore, it means that on the one hand, the first objective which is to compute $q(\theta | {y}\_{o})$ can be achieved without explicitly computing $q(x | {y}\_{o})$ but rather by augmenting the simulator with the noise model. On the other hand, the second objective which is to compute $q(x, z | {y}\_{o})$ can be achieved without computing $q(\theta | x)$. In my opinion, those two problems are hence orthogonal and it makes me wonder if combining the two actually has some additional value over considering those problems as disjoint.

---

> > > ### Author Response · Authors · 2022-08-03
> > > **Reply to clarifications**
> > >
> > > Thank you very much for the swift clarification. The problem that we are trying to solve occurs in the other direction: while NPE is not required to obtain the model misspecification probabilities, the analysis of model misspecification as well as the "denoising" is required to obtain reasonable estimates for the posterior, see for Example Figure 2, where NPE without our approach is unreliably overconfident. Hence, we would not view these parts as completely orthogonal.
> > > To be more precise, finding $q(x, z \mid {y}_o) \propto p({y}_o \mid x, z) q(x) p(z)$ as you mention is _a_ goal, but itself also a part of the other goal, which is to make NPE more reliable.
> > >
> > > > it makes me wonder if combining the two actually has some additional value over considering those problems as disjoint
> > >
> > > While it might be possible "in principle" to obtain $q(\theta | {y}_{o})$ using NPE directly after augmenting the simulator, there are several practical reasons for taking the approach we have, as we have listed here: https://openreview.net/forum?id=MHE27tjD8m3&noteId=aFwB3ExaXXN .
> > >
> > > Please let us know if you have any further questions or concerns. If you believe that our answers address all your concerns or you have any other thoughts on our approach, please let us know. All feedback is appreciated.

---

> > > > ### Comment · Reviewer_veSL · 2022-08-05
> > > > **Require further clarifications**
> > > >
> > > > > the analysis of model misspecification as well as the "denoising" is required to obtain reasonable estimates for the posterior, see for Example Figure 2, where NPE without our approach is unreliably overconfident.
> > > >
> > > > I agree that taking into account misspecification is important to avoid overconfident estimates. However, here I understand that you compare RNPE vs NPE without corrections but NPE with an augmented simulator does not appear in this Figure. Is this the case?
> > > >
> > > > Regarding the practical reasons for taking the approach of explicitly computing $q(x, z | {y}\_{o})$, I must admit that I do not get some of those.
> > > >
> > > > > Structure of the error model: Note that what we propose is a structural error model that allows for interpretation and critical assessment of model misspecification. Approaches using nuisance parameters commonly use a simpler error model, e.g. additive noise to extend the range of possible values.
> > > >
> > > > In my opinion, the error model used is decoupled from the way it is incorporated. You could as well augment the simulator with a complex noise model. Your method could maybe lead to better results but if it is the case, it should be demonstrated empirically.
> > > >
> > > > > Decoupling the simulator and the error model provides additional insight: By analyzing differences between observed data and simulated data the modeler can be aided in improving the simulation model. Knowing which summary statistics are misspecified and observing the effect on the distribution can be crucial for model development.
> > > >
> > > > As mentioned in my previous comment, computing $q(x, z | {y}\_{o})$ can be done without computing explicitly $q(\theta|x)$.
> > > >
> > > > > Including what is known to improve computational efficiency: Since we do have a distributional model for the error we can include this information in the inference procedure. The alternative would be to rely on NPE to learn the dependence on a potentially difficult heavy-tailed distribution (such as a Cauchy distribution), i.e. to recover an element of the simulator that is already known.
> > > >
> > > > As said above, I acknowledge that for a complex noise model, using the fact that the likelihood of the error model is actually tractable could lead to better performance. But if this is the core advantage of this method, it should be framed as is in the text and demonstrated empirically by comparing the performance against the simple approach of augmenting the simulator.
> > > >
> > > > > Black-box applicability: The error model can be integrated into existing black-box inference packages, such as SBI, and augment NPE. This allows practitioners access to interpretable insights into model misspecification without having to manipulate their code or alternatively to check a simulation model that is otherwise only available as black-box itself.
> > > >
> > > > In my opinion, integrating RNPE in SBI represents as much effort as using some probabilistic programming library to invert the error model.

---

> > > > > ### Author Response · Authors · 2022-08-06
> > > > > **Further clarifications**
> > > > >
> > > > > Thank you for following up. There seems to be miscommunication regarding the nature of the contribution or the issue around including the error model in the simulator. This post will hopefully help to mitigate this, but please continue to ask questions if the details remain unclear.
> > > > >
> > > > > > I agree that taking into account misspecification is important to avoid overconfident estimates. However, here I understand that you compare RNPE vs NPE without corrections but NPE with an augmented simulator does not appear in this Figure. Is this the case?
> > > > >
> > > > > That is indeed correct. We compare our error model to the standard NPE algorithm.
> > > > >
> > > > > >> Structure of the error model: Note that what we propose is a structural error model that allows for interpretation and critical assessment of model misspecification. Approaches using nuisance parameters commonly use a simpler error model, e.g. additive noise to extend the range of possible values.
> > > > >
> > > > > > In my opinion, the error model used is decoupled from the way it is incorporated. You could as well augment the simulator with a complex noise model.
> > > > >
> > > > > We agree with this statement. What we meant to point out is that one of the core contributions _is the error model_. However, you are correct in that there are indeed different approaches to estimate the quantities of interest. For this reason, the decoupling you mention is to some degree the reason we implemented the method as we did.
> > > > >
> > > > > To be more precise: we estimate $q(\theta \mid x)$, the simulator posterior, in the classical NPE sense. We can then define and and iterate on different noise models that allow for model criticism and a mechanism to correct the classical NPE approach. The model criticism algorithm will provide an estimate $q(x, z \mid y_o)$ and as part of that posterior estimates $P(z_i = 1 \mid y_o)$ for every summary statistic $i$. To draw samples from the posterior we now have _several choices_:
> > > > > - use samples from $q(x, z \mid y_o)$ for denoising, or,
> > > > > - select point estimates such as the maximum a posteriori (MAP) values. For example, we can set $z_i = 1$ if we have evidence that summary $i$ is misspecified or $z_i = 0$ if the evidence points towards a well specified summary statistic, resulting in a distribution $q(x\mid , z_1 = 1, z_2 = 0, \ldots, y_0)$, say. In the extreme case, where all statistics are found to be well-specified, we can forego the sampling step and simply use $q(\theta \mid x)$.
> > > > > Now, it would be possible, as you suggest, to rerun the simulator with a noise model under various assumptions (e.g. $z_i = 0$) and use NPE, but this entails _re-sampling_ and _re-estimating_ $q(\theta \mid y)$ every time we want to change the error distribution, i.e. if we wanted to obtain the posterior for $z_i = 0$ instead of a mixture.
> > > > > - Another alternative is to estimate $q(\theta \mid y_o, z)$, which is a variation of our approach as one would need to sample $z$ if one was interested in the marginal distribution $q(\theta \mid y_o)=\int q(\theta \mid y_o, z)q(z\mid y_o)dz$.
> > > > >
> > > > > > As mentioned in my previous comment, computing $q(x, z | {y}_{o})$ can be done without computing explicitly $q(\theta|x)$.
> > > > >
> > > > > That is correct and we belief this is advantageous. The fact that we can estimate $q(x, z \mid y_o)$ and $q(\theta \mid x)$ separately, means that we can change $q(x, z \mid y_o)$ to e.g. $q(x\mid z=0, y_o)$ while still being able to compute $q(\theta \mid y_o)$ without having to relearn $q(\theta \mid x)$. Similarly, the computation of (4) is straightforward if samples $\tilde{x} \sim q( x \mid y_o)$ are already available from the model criticism step.
> > > > >
> > > > > > Your method could maybe lead to better results but if it is the case, it should be demonstrated empirically.
> > > > >
> > > > > > But if this is the core advantage of this method, it should be framed as is in the text and demonstrated empirically by comparing the performance against the simple approach of augmenting the simulator.
> > > > >
> > > > > Perhaps this is the root of some misunderstanding: the argument of the article is how model criticism and a judicious choice of error model can help interpretability as well as posterior reliability. While we relied on several heuristics (see above) to decide on a particular implementation, the exact choice of algorithm (Monte Carlo average over denoising or flow) is not the main contribution and no claim regarding the efficiency of particular probabilistic algorithms to sample from the posterior in equation (3) is made in the paper.
> > > > >
> > > > > That said, we are happy to include this approach as an alternative implementation, discuss advantages and disadvantages and to provide simulation results.
> > > > >
> > > > > > In my opinion, integrating RNPE in SBI represents as much effort as using some probabilistic programming library to invert the error model.
> > > > >
> > > > > Our statement was not to suggest that integrating RNPE in SBI is easy. Rather we are saying that after the error model is integrated in SBI, it is easy for practitioners to use, even if they only have access to a black box simulator.

---

> > > > > > ### Comment · Reviewer_veSL · 2022-08-09
> > > > > > **Thanks for the clarifications**
> > > > > >
> > > > > > Thanks for the clarifications. I now understand that decoupling the error model from the parameter inference procedure can help iterate on this error model and facilitate the understanding of misspecified variables. I find that the paper would benefit from having this objective clearly written in place of the very abstract term "model criticism". For now, the contribution paragraph is
> > > > > >
> > > > > > > We propose robust neural posterior estimation (RNPE), an SBI framework that incorporates an error model, $p(y | x)$, to explicitly express our belief that a discrepancy might exist between the data generating process and the simulator. Following ideas from Box (1980), RNPE facilitates principled scientific modeling, allowing iteration between two complementary components: model criticism and robust inference.
> > > > > >
> > > > > > Although you mention this iteration between the two components, it reads as if your first contribution was to incorporate an error model. I would first start the paper by mentioning that incorporating the error model is important and that it could be done by incorporating it into the simulator. Then, the contribution paragraph would mention that you present a new framework that consists in decoupling the error model from the parameter inference, that it allows iterating over various error models efficiently and why this is useful.

---

> > > > > > > ### Author Response · Authors · 2022-08-09
> > > > > > > **Thank you for your valuable feedback**
> > > > > > >
> > > > > > > Thank you very much for your engagement and your valuable feedback. Your comments and the ensuing discussion have helped us to refine our arguments and ultimately to improve the quality of the paper.
> > > > > > >
> > > > > > > > Although you mention this iteration between the two components, it reads as if your first contribution was to incorporate an error model. I would first start the paper by mentioning that incorporating the error model is important and that it could be done by incorporating it into the simulator. Then, the contribution paragraph would mention that you present a new framework that consists in decoupling the error model from the parameter inference, that it allows iterating over various error models efficiently and why this is useful.
> > > > > > >
> > > > > > > Thank you for this suggestion. We have incorporated this feedback (and all the other feedback) into the revised version of the manuscript (which is now available) and our contribution section now follows the outline you propose.
> > > > > > >
> > > > > > > We hope the discussion and the changes to the manuscript address all or most of your concerns. If your view on the merits of our approach has changed after the discussion, we would be grateful if you could indicate this by adjusting the score.
> > > > > > >
> > > > > > > Thank you.
> > > > > > >
> > > > > > > The Authors

---

### Official Review · Reviewer_3EMZ · 2022-07-10

**Rating:** 3
**Confidence:** 3
**Soundness:** 2 fair
**Presentation:** 3 good
**Contribution:** 1 poor

**Summary:**

The authors proposed an extension of a class of simulation-based inference using neural networks called "robust neural posterior estimation". They intorduced an additional inference layer to a simulation-based inference method "neural posterior estimation" by combining it with an "error model" that is a generative model of data point from a simulated sample. In this work, the error model was specified by a spike-and-slab distribution, which in turn provides an indicator "posterior misspecification probability" for degrees of misspecification of each specified data summary statistics. Their experiments compared "robust neural posterior estimation" and "neural posterior estimation".


**Questions:**

- To me personally, there seems several sentences whose meanings were not fully clear and accurate throughout the paper in general. For example, on the sentence '... probabilistically “denoising” the observation, thus producing samples consistent with both the error model and simulator' in the page 3, it wasn't clear what "denosing" means here and what "consistent with both the error model and simulator" means exactly. For other example, the sentence 'Despite this, the general approach in SBI is ... to query it with the observed data, q(θ | x = yo)' seemed too coarse description of  the procedure; what would you mean by "query with" for example? I understand some of intensions behind are to make explanation intuiteve but it would be helpful to refine inaccurate explanations so that they sounds more accurate and rigorous.
- It would be helpful to add an exact literature that coined the term "neural posterior estimation" if there is any.
- It would be helpful to add an exact literature that used the term “simulation-to-reality gap” in the page 2.
- The explanation of the algorithm procedure below (3) does not read well. For example, it may be helpful to write them as a certain type of well-organised lists.

**Limitations:**

There would be no negative societal impact. The clear limitation is that there was no comparison with other existing methodologies apart from neural posterior estimation. The significance of the proposal would have to be justified clearly by either or both extensive experimental evaluation or comprehensive theoretical analysis.

**Strengths And Weaknesses:**

- If I understand correctly, a core of the proposed idea is to introduce an error model $p(y \mid x)$, a model that generates $y$ from a simulated sample $x \sim \text{Simulator}_{\theta}$. Would this approach can be equivalent with defining a new simulator by plugging the original simulator into some condition probability distribution: $p(y \mid x = \operatorname{Simulator}_\theta )$? In that case, other general methods in SBI or ABC would become available to learn all the parameters of the new simulator?
- In this work, a spike-and-slab distribution is used for the error model $p(y \mid x)$. The choise of the distribution seems to implicitly specify what type of misspecification we assume to happen because the model $p(y \mid x)$ describe how datum $y$ is generated from a simulation $x$. In spike-and-slab case, it seems like at first glance that an implicitly assumed type of misspefication is less frequent outlier. The performance of robust neural posterior estimation seems to be dependent of the choice of the error model to a large degree. The spike-and-slab would be the only choice that posterior misspecification probability can be computed? Would it be possible to provide other  available choices and examine impacts by them?
- Given that above perspectives, if it is possible to apply other other general methods in SBI or ABC to learn the entire parameter of the simulator, there would be vairous other robust inference methods available such as one based on maximum mean descrepancy. If this is the case, it would be helpful to see or commnet on how efficient the proposed procedure is than others. In this respect, the reviewer was personally wondering if the broad name "robust" neural posterior estimation is throughly accurate/appropriate to represents the exact content of the proposal, given that there seems to exist many possible approaches to "robust" inference.
- The experiments seem to compare "robust neural posterior estimation" with "neural posterior estimation" only. It would be strongly helpful to compare it with other existing methodologies such as ones cited in Section 3.2 and to clarify when and how the robust neural posterior estimation has the advantage to others. The performances of robust neural posterior estimation and neural posterior estimation would also be dependent of the choice of neural networks to be used, where therefore it may also be helpful to justify that the used models were reasonable somewhere in the main paper.

---

> ### Author Response · Authors · 2022-08-02
> **Thank you for your review.**
>
>
> Thanks for your review and thank you for highlighting the sections you found unclear, we will amend these sections for the final draft. We hope that the below can address some of your concerns.
>
> > Would this approach can be equivalent with defining a new simulator by plugging the original simulator into some condition probability distribution?
>
> We appreciate this question. These approaches are definitely related. However, using RNPE with an explicit error model as we do here has the following advantages:
>
> 1. Structure of the error model: Note that what we propose is a structural error model that allows for interpretation and critical assessment of model misspecification. Approaches using nuisance parameters commonly use a simpler error model, e.g. additive noise to extend the range of possible values.
> 2. Decoupling the simulator and the error model provides additional insight: By analysing differences between observed data $y$ and simulated data $x$ the modeller can be aided in improving the simulation model. Knowing which summary statistics are misspecified and observing the effect on the distribution can be crucial for model development.
> 3. Including what is known to improve computational efficiency: Since we do have a distributional model for the error we can include this information in the inference procedure. The alternative would be to rely on NPE to learn the dependence on a potentially difficult heavy tailed distribution (such as a Cauchy distribution), i.e. to recover an element of the simulator that is already known.
> 4. Black-box applicability: The error model can be integrated into existing black-box inference packages, such as SBI, and augment NPE. This allows practitioners access to interpretable insights into model misspecification without having to manipulate their code or alternatively to check a simulation model that is otherwise only available as black-box itself.
>
> It is evident that we could have been clearer when justifying the approach used, we will amend this in the final draft.
>
>
> > The choise of the distribution seems to implicitly specify what type of misspecification we assume to happen because the model  describe how datum  is generated from a simulation .
>
> You are correct that the choice of the distribution does to some extent specify what types of misspecification we assume will happen. However, error model parameters can be inferred (as we do for the misspecification probabilities), such that you get a posterior distribution over the error model parameters. Frequently, the lack of i.i.d. data in SBI makes more sophisticated error models challenging. We note that in the more general SBI setting, practitioners are still forced to make assumptions on the model error - i.e. often that there is none - and as demonstrated here, this assumption leads to misleading results when violated.
>
> > The performances of robust neural posterior estimation and neural posterior estimation would also be dependent of the choice of neural networks to be used, where therefore it may also be helpful to justify that the used models were reasonable somewhere in the main paper.
>
> We do briefly justify the architecture choices in the setup section starting on line 226, but leave more detailed hyperparameter choices to the appendix as the choice relies on a deeper understanding of the specific algorithms (neural spline flows and block neural autoregressive flows). Note that the same neural spline flow architecture was used in both RNPE and NPE, which ensures the results are comparable.
>
> > It would be helpful to add an exact literature that coined the term "neural posterior estimation" if there is any.
>
> To our knowledge, this term was introduced in [1] in the form of Sequential Neural Posterior Estimation. We will add that reference.
>
> > It would be helpful to add an exact literature that used the term “simulation-to-reality gap” in the page 2.
>
> This and similar terms have been used for a long time to highlight problems that arise because simulations inadequately describe reality. Unfortunately, we are not aware of the precise origin.
>
> > The explanation of the algorithm procedure below (3) does not read well. For example, it may be helpful to write them as a certain type of well-organised lists.
>
> The algorithm is available in pseudo-code in Algorithm 1 after line 126, which is more like a list. Perhaps this serves as a better explanation?
>
> Please let us know if you have any further questions.
>
> [1] Lueckmann, J. M., Goncalves, P. J., Bassetto, G., Öcal, K., Nonnenmacher, M., & Macke, J. H. (2017). Flexible statistical inference for mechanistic models of neural dynamics. Advances in neural information processing systems, 30.

---

> > ### Comment · Reviewer_3EMZ · 2022-08-08
> > **Response**
> >
> > Thank you for your detailed answer. I understand that robustness and interpretability in SBI are important challenges, but in summary I am personally still not convinced on that novelity/significance of the proposed approach so far, and therefore my score has not been changed.
> >
> > In my current understanding, the acutal methodological development is as follows:
> >
> > 1. overlay a spike-and-slap distribution model on a given simulator-model
> > 2. estimate the parameter of both the models
> > 3. use the mixture-proportion parameter of the spike-and-slap distribution model as an interpretable indicator of a simulator-model error.
> >
> > Since the idea looks relatively simple (simplicity is good given a nice justificaiton), I personally would like to see the clear and rigorous justificaiton underpinned by either theoreticla analysis or empirical study that reasonablly claims that the proposal would be better than other existing approahces in some scenario of focus. For example,
> >
> > - One may use existing robust ABC (e.g. MMD-ABC) for a given simulator-model and afterwards perform post-hoc model check such as postrior-predictive checks rather than using the proposed built-in error model, which would also be interpretable. What'd be wrong in this case? What property of the proposed approach is nicer than such an approach exactly?
> > - What does looking into the mixture-proportion parameter of the spike-and-slap distribution means statistically? The mixture-proportion looks like a certain indicator on surface but could we actually trust the value as a magnitude of model error in some scenarios? And if so what kind of scenario would it be? Would using the spike-and-slap distribution makes any fundamental difference from using just a Gaussian distribution and checking the estimated variance alternatively?
> >
> >
> >
> > > You are correct that the choice of the distribution does to some extent specify what types of misspecification we assume will happen. However, error model parameters can be inferred (as we do for the misspecification probabilities), such that you get a posterior distribution over the error model parameters.
> >
> > I suppose the choice of a "error model" specify our prior assumption on misspecification type exactly rather than "to some extent". While the parameter is learnable indeed, it finds the best-fit error model within a user-specified class (i.e. error type hypothesis). Use of the error model or similar approach in general seems inseprarable from implicit assumption of misspecification type. It seems worth clarifying what misspecification type to be focused in this paper (i.e. investigating and clarifying when the spike-and-slap approach would most likely be successful since it is the only and main error model used in the paper).

---

> > > ### Author Response · Authors · 2022-08-08
> > > **Clarifications**
> > >
> > > Thank you for your feedback. We have collected some answers for your remaining concerns below.
> > >
> > > > In my current understanding, the acutal methodological development is as follows:
> > > > - overlay a spike-and-slap distribution model on a given simulator-model
> > > > - estimate the parameter of both the models
> > > > - use the mixture-proportion parameter of the spike-and-slap distribution model as an interpretable indicator of a simulator-model error.
> > >
> > > That is correct. __In addition__, one can use the Monte Carlo samples from $p(x | y)$ in the _standard NPE posterior_ to obtain samples from $q(\theta | y)$, the noise corrected posterior.
> > >
> > > > One may use existing robust ABC (e.g. MMD-ABC) for a given simulator-model and afterwards perform post-hoc model check such as postrior-predictive checks rather than using the proposed built-in error model, which would also be interpretable. What'd be wrong in this case? What property of the proposed approach is nicer than such an approach exactly?
> > >
> > > That's an interesting question. There are several points that make our approach a bit more appealing:
> > > 1. Instead of posterior, we are comparing to the prior predictive distribution, which itself is naturally a more robust approach - i.e. if the posterior predictive looks bad, how does one know if this is due to failure of the inference procedure, or limitations of the simulator?
> > > 2. Our approach works without running inference first: we do not need to obtain an approximate posterior like ABC to check whether the model is misspecified. Using ABC requires carrying out the inference part (finding the posterior) first.
> > > 3. It works together with neural posterior estimation: we can adjust amortized neural posterior estimation techniques by using the samples from the (denoised) error model (as mentioned above). Neural posterior estimation usually requires less samples and is more accurate than ABC [3].
> > >
> > > > What does looking into the mixture-proportion parameter of the spike-and-slap distribution means statistically? The mixture-proportion looks like a certain indicator on surface but could we actually trust the value as a magnitude of model error in some scenarios? And if so what kind of scenario would it be?
> > >
> > > The difference between prior $p(z = 1)$ and (estimated) posterior probabilities, $p(z = 1 | y)$, allow us to rely on the scientific evidence frameworks of Bayesian epistemology [1, 2]. This field of philosophy argues that $p(z = 1 | y) > p(z = 1)$ provides _evidence_ for the hypothesis that $z = 1$. Similarly, $ p(z = 1 | y) < p(z = 1)$ yields evidence against the hypothesis.
> > >
> > > > Would using the spike-and-slap distribution makes any fundamental difference from using just a Gaussian distribution and checking the estimated variance alternatively?
> > >
> > > One advantage of our approach is that one could have a small discrepancy in the data space, i.e. $y$ vs $x$, that can be detected well by the RNPE, in which case the "magnitude" is small, but p(z=1|y) will be very high.
> > >
> > >
> > > [1] Hartmann, Stephan; Sprenger, Jan (2010). "Bayesian Epistemology". The Routledge Companion to Epistemology. London: Routledge. pp. 609–620.
> > >
> > > [2] Talbott, William, "Bayesian Epistemology", The Stanford Encyclopedia of Philosophy (Winter 2016 Edition), Edward N. Zalta (ed.), URL = <https://plato.stanford.edu/archives/win2016/entries/epistemology-bayesian/>.
> > >
> > > [3] Lueckmann, Jan-Matthis, et al. "Benchmarking simulation-based inference." International Conference on Artificial Intelligence and Statistics. PMLR, 2021.

---

> > > > ### Author Response · Authors · 2022-08-08
> > > > **Further clarifications**
> > > >
> > > > > I suppose the choice of a "error model" specify our prior assumption on misspecification type exactly rather than "to some extent". While the parameter is learnable indeed, it finds the best-fit error model within a user-specified class (i.e. error type hypothesis). Use of the error model or similar approach in general seems inseprarable from implicit assumption of misspecification type. It seems worth clarifying what misspecification type to be focused in this paper (i.e. investigating and clarifying when the spike-and-slap approach would most likely be successful since it is the only and main error model used in the paper).
> > > >
> > > > We use the terminology "to some extent" to say that the very heavy tail in the error model is able to accommodate a large number of error sources, regardless of their true distribution. To elaborate on this point: if an observation falls into the support of the simulator, our algorithm will identify the observation to come from the spike with high probability. However, an observation taking values far outside the support will adversely affect the posterior estimate. Our algorithm will assign high probability to the slab, which adds a heavy tailed noise model on top of the simulator and hence assigns the observation high probability also. In this way, the observation is no longer an outlier with a strong effect on the posterior.
> > > >
> > > > It is important to point out that due to the heavy-tailed nature, this approach will yield reasonable results for a large range of potential (true) error distributions. The reason for this is that in simulation-based inference, we commonly observe _one_ sample of true data $y_o$ and therefore the behaviour for this _particular_ $y_o$ under the error model is important, not it's distribution [F].
> > > > The use of heavy-tailed distributions for the mitigation of outlier influence has a long history in statistics, see e.g. the review in [4] and references therein.
> > > >
> > > > [4] O’Hagan, Anthony, and Luis Pericchi. "Bayesian heavy-tailed models and conflict resolution: A review." Brazilian Journal of Probability and Statistics 26.4 (2012): 372-401.
> > > >
> > > > [F] Footnote: If we only have one sample, defining and estimating distributions is difficult, a problem also inherent to the MMD approach which is defined for a sample size of one, but might not yield a very accurate estimate of the true MMD.

---

### Official Review · Reviewer_HK3N · 2022-07-11

**Rating:** 6
**Confidence:** 5
**Soundness:** 3 good
**Presentation:** 3 good
**Contribution:** 3 good

**Summary:**

Simulation-based inference (SBI) enables Bayesian parameter inference in complex scientific simulators for which the evaluation of the likelihood is usually intractable. In most recent works on SBI this is achieved by conditional density estimation with neural networks trained on data simulated from the model. However, SBI generally assumes that the model is well-specified, and up to now it is not clear how SBI methods behave in the misspecified case, e.g., when the data distribution induced by the simulator (and the prior) does not contain the experimentally observed data.
This paper proposes a new SBI approach, RNPE, to enable inference even in the misspecified scenario, and to additionally obtain insights into the nature of misspecification present in the model. The goal is to obtain the posterior over the model parameters given the experimentally observed data, $p(\theta | y)$, given only (potentially misspecified) simulations from the model $(\theta, x)$. RNPE (robust NPE) extends the established SBI method Neural Posterior Estimation (NPE) to a two-step procedure: First, one performs neural density estimation on simulated data to learn $p(x)$, and assumes a noise model of the mismatch between the observed data $y$ and the simulated data $x$ , $p(y | x)$, to then obtain “denoised” posterior samples from $p(x|y)$ via MCMC. Second, one performs NPE on simulated data to obtain an amortized posterior estimator $q(\theta | x)$ . Finally, one can obtain the posterior via a Monte Carlo approximation by repeatedly sampling denoised $\tilde{x}$ from the first step, and obtaining corresponding posterior samples from $q(\theta | \tilde{x})$.
RNPE is compared to NPE on three misspecified SBI example problems and shows consistently better performance on the evaluated metrics. In addition, the authors demonstrate that the inferred noise model from the first step can by useful to diagnose whether specific summary statistics are misspecified, which in turn can serve as a tool for model criticisms.

**Questions:**

3. **Questions:**
As explained in detail above, I think it is essential to evaluate RNPE performance on a fully tractable example like the Gaussian task, in order to answer questions like:
    - Are RNPE and NPE empirically equivalent in the well-specified case?
    - How does slowly increasing the amount of misspecification change metrics like posterior mean error, variance error, dispersion, C2ST between the analytical and the NPE and RNPE posteriors?
    - How do the posterior predictive distributions of the true, NPE and RNPE posteriors look like in the misspecified case?
    - How does the Monte Carlo estimate in RNPE affect the RNPE vs the true posterior?

    Evaluation of additional metrics is especially important as in my impression, the expected coverage metric can be misleading. For example, in Figure 2b SIR it seems clear that NPE is massively overconfident, suggestion that it is too small posterior variance. But what is actually happening is that the NPE posterior is highly biased to other regions in the parameters space and actually has a very large variance. This behaviour would be immediately evident from a simulation-based calibration visualization of the ranks (Talts et al. 2018). I see that it might be challenging to perform SBC over $y$, but there might alternative metrics like the local coverage test (Zhao et al.). Alternatively, one could obtain reference posteriors for the SIR (Lueckmann et al. 2021) and then calculate metric between the posterior directly to evaluate RNPE vs NPE performance.

    The results of the spike-and-slap error model are promising, however, I am wondering whether you investigated other error model, and if so, how did they compare?

    The denoised data distributions shown in Figure 4b and in the appendix tend to have quite funky shapes. Are there intuitive explanations why this is the case, e.g., in 4b, what is the reason for the bimodal shape, opposed to the unimodal shape of the simulated data? That makes me wonder, how was the training of the flow and successful approximation of $p(x)$ evaluated?

    In the context of Figure 4c you point out that in contrast NPE, RNPE puts more posterior mass around the underlying ground-truth parameters. Here, I think it is important to note that in general posterior do not have be centered on ground-truth parameters, so in principle NPE could be correct here as well. One way to tell them apart would be checking the posterior predictive samples—how do they look like?

**Limitations:**

Limitations of RNPE to data and parameter space dimensionality of the SBI problem should be discussed in more detail (see above). Overall, I think this is an important and promising contribution that requires some additional evaluation and discussion of limitations.

**Strengths And Weaknesses:**

- *Originality:*
The paper introduces a new SBI method, RNPE which extends the established SBI method neural posterior estimation (NPE) to handle scenarios where the assumed model is misspecified. This is a new direction of research in the field of SBI and RNPE clearly differs from previously proposed approaches in this direction. Most of the previous work is adequately cited, except for the work on neural ratio estimation for SBI (NRE, Hermans et al. 2019, Durkan et al. 2019, Thomas et al. 2020), which should be seen as the third arm of neural network-based SBI methods (next to NPE, and NLE). Additionally, I think it should be considered citing the original papers on HMC alongside the mixed HMC paper, e.g., Neal (1995), MacKay (2003), or Neal (2011).
- *Quality*:
The idea of RNPE to assume a noise model and to learn $p(x)$ via density estimation to then obtain denoised data, followed by NPE, seems quite original, complete and overall technically sound to me. The required derivations are straight forward to presented clearly in the paper. I have two critique points regarding the evaluation:
    - first, I think it is essential to fully study and evaluate RNPE on a tractable example like the proposed Gaussian task (the Gaussian task is ideal because one can gradually increase the misspecification from well-specified (equal variance) to strongly misspecified (variance, say 25), and because one has easy access to the ground-truth analytical posterior).
    In particular, it would good to first, empirically demonstrate that RNPE and NPE are equivalent in the well-specified case, and second, it would be good to show how the Monte Carlo (MC) approximation in the last step affects the results. For example, it would good to see (in the misspecified case) how as a function of the number of MC samples, the RNPE posterior samples approach the true posterior.
    - second, I think it is essential to use more evaluation metrics for all three tasks. On the Gaussian task with access to reference posteriors one could additionally show the error in posterior mean, in posterior variance, and the actual dispersion between true and approximated posteriors. For all three task one could show additional calibration metrics, as I believe the expected posterior coverage can be misleading in some scenarios (see questions below). One possibility would be simulation-based calibration (SBC, Talts et al. 2018), although I acknowledge that it might be computationally infeasible because the entire RNPE procedure would need to repeated ~100 for different $y$. Alternatively, local coverage tests around the observed $y$ could be performed using Zhao et al. 2021 ([https://proceedings.mlr.press/v161/zhao21b.html](https://proceedings.mlr.press/v161/zhao21b.html)).
    Finally, I think it would be instructive to additionally compare posterior predictive samples of the different approaches, e.g., by sampling parameters from the posterior and then passing them through simulator and the noise model and compare “simulated” and “noised” predictive samples.

    The evaluation on three example tasks is informative. However, all three tasks are rather low-dimensional (or designed to be) in the data and the parameter domain. I am wondering how RNPE scales to higher-dimensional problems. NPE was shown to infer plausible posteriors in up to 30-dimensional parameters spaces (Goncalves et al. 2020, eLife) and large data spaces in the image domain using embedding nets (see NPE results in the GATSBI paper by Ramesh et al. 2021), but I fear that this will be challenging for RNPE for several reasons. RNPE requires a Monte Carlo approximation from NPE, which might become computational expensive in large parameter spaces. RNPE requires learning $p(x)$, which will quickly become challenging as $x$ is not reduced to low-dimensional summary statistics.
    For the first problem I think it is feasible to design another evaluation task with more parameters, e.g., a high-dimensional Gaussian task, and to show Monte Carlo approximation behaves. For the second problem I see that it is challenging to extend RNPE to be used with embedding networks. But even more importantly, I think it is essential that the reader is informed about these limitations, which does not seem to be the case in the current version of the paper.

- *Clarity*:
The paper is clearly written and well organized. Limitations should be stated more clearly (see above). Nitpick: the numbers in 227 and 228 are formatted inconsistently.
- *Significance:*
This is a significant contribution. RNPE appears conceptually solid and shows promising results. Once evaluated thoroughly, I believe it has the potential to be used both by SBI practitioners (depending on the availability of RNPE in an open-source package, e.g., $\texttt{sbi}$ Tejero-Cantero et al. 2020), and for future research on new SBI methods.

---

> ### Author Response · Authors · 2022-08-02
> **Thank you for your review.**
>
> Thank you for your review and thoughtful suggestions.
>
> > Are RNPE and NPE empirically equivalent in the well-specified case?
>
> We have now included results for the well-specified case (currently in appendix B, although we will–space permitting–work this into the main paper for the final version). Note that despite the sparsity promoting error model, we do still expect somewhat conservative results when using an error model in the well-specified case. It is worth noting, that when using RNPE, you also have access to the NPE posterior at no additional cost, so if the evidence for the model being well-specified is strong, we could choose to collapse RNPE down to NPE, with the reassurance of empirical evidence against substantial misspecification.
>
>
> > How does slowly increasing the amount of misspecification change metrics like posterior mean error, variance error, dispersion, C2ST between the analytical and the NPE and RNPE posteriors?
>
> We have included an example of the Gaussian task in which we gradually increase misspecification and plot the resulting posteriors (for now we have added an example plot in appendix B.). Given the relatively short turnaround, we have not been able to also add additional metrics such as C2ST, but those will be added to the final version for tractable examples.
>
>
> > The denoised data distributions shown in Figure 4b and in the appendix tend to have quite funky shapes. Are there intuitive explanations why this is the case, e.g., in 4b, what is the reason for the bimodal shape, opposed to the unimodal shape of the simulated data?
>
> The bimodality is caused by the shift between being in the spike (the error model encouraging $\tilde{x} \approx y_o$, for each summary statistic) and the slab.
>
> > How does the Monte Carlo estimate in RNPE affect the RNPE vs the true posterior?
>
> In order to correctly address your question, can we ask: Is your concern that the Monte Carlo estimate of $p(\theta|y)$ (shown in equation 4) could become high variance? To investigate this it is likely easiest to use a task where $p(x|y)$ and $p(\theta|x)$ are known (e.g. use a Gaussian prior, likelihood and error model), so we can eliminate errors introduced by the flow approximations, and cheaply investigate the effect of dimensionality on the Monte Carlo error. We can investigate this in the final version.
>
> Thank you for your remarks regarding including citations for neural ratio estimation and HMC. We will be sure to amend this for the final draft.

---

> > ### Comment · Reviewer_HK3N · 2022-08-08
> > **Concerns remain: discussion of limitations & comparison to other approaches**
> >
> > Thank you for conducting additional experiments for the well-specified case, and for increasing level of misspecification in the Gaussian task. The results answer most of my questions.
> >
> > However, as pointed out in my initial review, I think it is essential that the limitations of RNPE are discussed in detail. It seems this is not the case in the updated version of the submission. In particular I am concerned about :
> > - the computational limitations of RNPE: in addition to performing NPE, one has to learn $p(x)$ and then infer $p(x | y)$ using e.g., MCMC. This works fine for the low-dimensional examples presented here, but it should be discussed how they limit the application of RNPE to higher-dimensional problems.
> > - the choice of the error model: it should be discussed how the choice of error can affect the results. In your examples the models were misspecified only in one data dimension. How could one deal with a potential correlation between misspecified data dimensions?
> >
> > Additionally, having read the comments by the other reviewers I have to agree with the concern of a missing comparison to other approaches towards robust SBI, most prominently, Frazier et al. 2020 who build the error model into the simulator. It should be straight forward to do this using NPE (much more efficiently than with ABC). The advantages (efficiency?) and disadvantages (interpretability) between this approach and NPE should studied empirically, or at least discussed in detail.

---

> > > ### Author Response · Authors · 2022-08-09
> > > **Thank you for the feedback**
> > >
> > > Thank you very much for your feedback. We have incorporated your feedback in a revised version of the manuscript.
> > >
> > > > the computational limitations of RNPE: in addition to performing NPE, one has to learn  and then infer  using e.g., MCMC. This works fine for the low-dimensional examples presented here, but it should be discussed how they limit the application of RNPE to higher-dimensional problems.
> > >
> > > We agree with this and we have added a discussion of those limitations to our conclusion. We believe that the dimensionality is one of the smaller problems, as our approach relies on usually handcrafted summary statistics for interpretability. The number of handcrafted summary statistics is usually not in the orders of magnitude where algorithms like HMC fail. Nevertheless, the general computational overhead of MCMC is a limitation that we added.
> > >
> > > > the choice of the error model: it should be discussed how the choice of error can affect the results. In your examples the models were misspecified only in one data dimension. How could one deal with a potential correlation between misspecified data dimensions?
> > >
> > > We agree with this and discuss this here also: https://openreview.net/forum?id=MHE27tjD8m3&noteId=m8nhERpyOZP
> > >
> > > We will add a discussion and we will include a task in which all the summary statistics are equally misspecified (e.g. a model with additive noise) and investigate the performance in this case i.e. where the “sparsity” assumption does not hold.
> > >
> > > > Additionally, having read the comments by the other reviewers I have to agree with the concern of a missing comparison to other approaches towards robust SBI, most prominently, Frazier et al. 2020 who build the error model into the simulator. It should be straight forward to do this using NPE (much more efficiently than with ABC). The advantages (efficiency?) and disadvantages (interpretability) between this approach and NPE should studied empirically, or at least discussed in detail.
> > >
> > > We have included additional experiments using our proposed error model _inside_ the simulator in conjunction with NPE. We refer to this approach as NNPE in the paper. The results are as expected: Since NNPE and RNPE are targeting the same distribution, both lead to qualitatively similar results. However, the fact that the error model is known analytically enables inference to be carried out slightly more efficiently using RNPE. It is also important to point out that RNPE comes with several other advantages as discussed here: https://openreview.net/forum?id=MHE27tjD8m3&noteId=hpsqQ686Frpc

---

> > > ### Author Response · Authors · 2022-08-09
> > > **Addressing concerns**
> > >
> > > Thank you for your comment.
> > >
> > > Regarding scaling to higher dimensional problems, due to the tight turnaround, we did not have time to include a higher dimensional task before the end of the discussion period. However, one thing we would like to note is that the individual components of RNPE (HMC and normalising flows) are both known to perform well for high dimensional tasks (e.g. for normalising flows see the RealNVP paper by Dinh et al., 2017 or Lueckmann et al., 2021).

---

### Official Review · Reviewer_yiXg · 2022-07-11

**Rating:** 7
**Confidence:** 4
**Soundness:** 3 good
**Presentation:** 3 good
**Contribution:** 3 good

**Summary:**

This study develops a new method for simulation-based inference (RNPE), capable of diagnosing model misspecification and reducing its impact on posterior inference. The empirical results on three tasks show that RNPE performs more reliable inference than a previous method (neural posterior estimation, NPE) according to a few metrics -- log probability of true parameters under the inferred posterior, and posterior coverage. In addition, RNPE provides a misspecification probability for each summary statistic on each task, information that NPE does not provide.


**Questions:**

As written above, the manuscript would benefit if the authors would include a task with misspecification across several summary statistics.


**Limitations:**

The authors have adequately addressed the limitations of their work.


**Strengths And Weaknesses:**

### Originality

Simulation-based inference has the potential to accelerate the process of scientific discovery across domains of science. One of the main limitations of this set of tools is the problem of model misspecification, a problem which has rarely been tackled. The questions addressed by this study are thus timely and relevant.


### Quality

The paper is technically sound, claims are backed by empirical evidence, and the contributions are appropriately put in the context of previous work. However, I have one point that I believe would be crucial for the authors to tackle:

-RNPE was tested in tasks for which the misspecification impacted only one summary statistic. Although this could be a realistic scenario in some cases, for many cases, the misspecification is across several summary statistics. In such cases, and if the summary statistics are not correlated with each other, then the extension to misspecification across several summary statistics seems trivial. However, in many realistic cases, there could be some residual correlation between summary statistics, and there one might expect the feedback from RNPE regarding misspecification probabilities to be less precise: there could be some indeterminacy in terms of misspecification, e.g. with a trade-off between misspecifications of different summary statistics. I believe the study would benefit from discussing such cases and including at least one task tackling this.


### Clarity

The manuscript is clearly written, providing enough information to understand the technical contribution and empirical results. I just have one point that I would appreciate the authors to clarify:

-in Figure 3b, the density for the denoised samples is shifted towards the right, without (seemingly) any mass around the observation. I might be misunderstanding something, but wouldn't one expect some non-negligible mass around the observation? If so, is it hidden in the plot by the green lines, and could one make it more visible?



### Significance

The technical development and empirical results will be of interest to the ML community. In addition, the technical development could quite effortlessly be extended to other simulation-based inference algorithms, a possibility that will certainly entice other researchers on this topic.

---

> ### Author Response · Authors · 2022-08-02
> **Thank you for your review.**
>
> Thank you very much for the review and the positive feedback and thank you for appreciating the timeliness and relevance of the problem we consider.
>
> > In Figure 3b, the density for the denoised samples is shifted towards the right, without (seemingly) any mass around the observation. I might be misunderstanding something, but wouldn't one expect some non-negligible mass around the observation?
>
> This is an interesting question. There does not necessarily have to be significant mass around the observation. The denoised samples are samples approximately from $p(x|y_o) \propto p(y_o|x)p(x)$. E.g. consider a one dimensional example, where the observation falls outside the support of the simulated data. In this case in the region surrounding the observation in $p(x|y_o) \propto p(y_o|x)p(x)$ will have negligible mass (as the $p(x)$ will be approximately zero). Note we are showing the most misspecified summary statistic in this plot - for the other statistics we see significant mass on the observed value.
>
> > As written above, the manuscript would benefit if the authors would include a task with misspecification across several summary statistics.
>
> Although the misspecification in our examples primarily impacts single summary statistics, other summary statistics are affected to some extent on all the tasks. Nevertheless, we agree with the usefulness of such an example and we will include a task in which all the summary statistics are equally misspecified (e.g. a model with additive noise) and investigate the performance in this case i.e. where the “sparsity” assumption does not hold.
>
> > There could be some indeterminacy in terms of misspecification, e.g. with a trade-off between misspecifications of different summary statistics.
>
> You are correct that there can be "indeterminacy" between different summary statistics. Ideally, one would look at and interpret all the misspecification probabilities jointly rather than marginally, which would give insight into these effects. Obviously this becomes challenging beyond e.g. pairwise interactions. We will add this as a discussion point in the final manuscript.

---

### Author Response · Authors · 2022-08-09
**Update of manuscript.**

Thank you all for your thoughtful reviews and discussion. We have updated the manuscript with the changed sections highlighted in red. The changes include (but are not limited to):

1. Making it clear that adding error into the simulator is an alternative approach, and discussing its limitations. We term the approach of incorporating the error in the simulator and using NPE as Noisy NPE (NNPE)
2. Including comparison to NNPE in the simulator parameter inference performance results (log probability $\theta^*$, coverage properties and MSE to point estimates in the appendix).
3. Highlighting the fact that the model criticism component can be decoupled from the inference component.
4. Better addressing limitations in the conclusion, such as the additional computational cost of RNPE over NPE, and only investigating a single error model choice.
5. Including a plot of the posterior predictive distribution for the SIR example in the appendix (corresponding to the posteriors in Fig 3c).

Thanks again for all your comments, and please consider reviewing the updated manuscript to see if it has addressed any of your concerns.

---

### Meta-Review · Area_Chair_94A4 · 2022-08-20

**Recommendation:** Accept
**Confidence:** Less certain

**Metareview:**

The reviewers, based on their scores, are not in consensus about this paper. However, there has been an extensive amount of conversation between reviewers, amongst themselves and with the authors, who have made significant updates to the manuscript that have been appreciated by the reviewers. Two reviewers champion the paper, while another reviewer maintained his/her score based on the original version of the manuscript, upon reading the updated version, only voices a mild concern around significance. The final reviewer who voted negatively simply presents novelty concerns.

After a careful read of the manuscript myself, I would like to break the vote towards acceptance. This is a topic that is often overlooked in this field and I think that the authors present an interesting approach and, perhaps more importantly, that this will lead to interesting discussion and further work in this field.

I appreciate the authors' careful attention to revising their manuscript and thank the reviewers for their balanced approach to reviewing and discussing this paper. I hope that the authors address any remaining concerns in preparing the final version of their manuscript,

**Award:**

No

---

### Decision · Program_Chairs · 2022-09-14

Accept